# A Topological Perspective on Causal Inference

**Duligur Ibeling**
Department of Computer Science
Stanford University
duligur@stanford.edu

**Thomas Icard**
Department of Philosophy
Stanford University
icard@stanford.edu

## Abstract

This paper presents a topological learning-theoretic perspective on causal inference by introducing a series of topologies defined on general spaces of structural causal models (SCMs). As an illustration of the framework we prove a topological causal hierarchy theorem, showing that substantive assumption-free causal inference is possible only in a meager set of SCMs. Thanks to a known correspondence between open sets in the weak topology and statistically verifiable hypotheses, our results show that inductive assumptions sufficient to license valid causal inferences are statistically unverifiable in principle. Similar to no-free-lunch theorems for statistical inference, the present results clarify the inevitability of substantial assumptions for causal inference. An additional benefit of our topological approach is that it easily accommodates SCMs with infinitely many variables. We finally suggest that the framework may be helpful for the positive project of exploring and assessing alternative causal-inductive assumptions.

## 1   Introduction and Motivation

In the background of any investigation into learning algorithms are no-free-lunch phenomena: roughly, the observation that assumption-free statistical learning is infeasible in general (see, e.g., [33, Ch. 5] for a formal statement). Common wisdom is that learning algorithms and architectures must adequately reflect non-trivial features of the data-generating distribution to gain inductive purchase.

For many purposes we need to move beyond passive observation, focusing instead on what *would* happen were we to *act* upon a given system. Even further, we sometimes desire to *explain* the behavior of a system, raising questions about what *would have* occurred had some aspects of a situation been different. Such questions depend not just on the data distribution; they depend on deeper features of underlying data-generating *processes* or *mechanisms*. It is thus generally acknowledged that stronger assumptions are required if we want to draw *causal* conclusions from data [35, 28, 20, 30, 32].

Whether implicit or explicit, any approach to causal inference involves a space of candidate causal models, viz. data-generating processes. Indeed, a blunt way of incorporating inductive bias is simply to *omit* some class of possible causal hypotheses from consideration. Many (im)possibility results in the literature can accordingly be understood as pertaining to all models within a class. For instance, if we can restrict attention to *Markovian* models that satisfy *faithfulness*, then we can *always* identify the structure of a model from experimental data (e.g., [11, 35]). If we can restrict attention to Markovian (continuous) models with linear functions and *non*-Gaussian noise, then *every* model can be learned even from purely observational data [34]. As a negative example, in the larger class of (not necessarily Markovian) models, *no* model can *ever* be determined from observational data alone [35, 2].

At the same time, in many settings it is sensible to aim for results with "nearly universal" force. It is natural to ask, e.g., within the class of all Markovian models, how "typical" are those in which the faithfulness condition is violated? This might tell us, for instance, how typically we could expect failure of a method that depended on these assumptions. A well-known result shows that, fixing

35th Conference on Neural Information Processing Systems (NeurIPS 2021).

any particular causal dependence graph, such violations have *measure zero* for any smooth (e.g., Lebesgue) measure on the parameter space of distributions consistent with that graph [24]. In fact, the standard notion of statistical consistency itself, which underlies many possibility results in causal inference, requires omission of some purportedly "negligible" set of possible data streams [9, 35].

There are two standard mathematical approaches to making concepts like "typical" and "negligible" rigorous: *measure-theoretic* and *topological*. While the two approaches often agree, they capture slightly different intuitions [25]. One virtue of the measure-theoretic approach is its natural probabilistic interpretation: intuitively, we are exceedingly *unlikely* to hit upon a set with measure zero. At the same time, the measure-theoretic approach is sometimes criticized in statistical settings for its alleged dependence on a measure, and this has been argued to favor topological approaches (see, e.g., [3] on no-free-lunch theorems). The latter of course in turn demands an appropriate topology.

In the present work we show how to define a sequence of meaningful topologies on the space of causal models, each corresponding to a progressively coarser level of the so called *causal hierarchy* ([29, 2]; see Fig. 1 for an abbreviated pictorial summary). We aim to demonstrate that topologizing causal models in this way helps clarify the scope and limits of causal inference under different assumptions, as well as the potential empirical status of those very assumptions, in a highly general setting.

Our starting point is a canonical topology on the space of Borel probability distributions called the *weak topology*. The weak topology is grounded in the fundamental notion of *weak convergence* of probability distributions [4] and is thereby closely related to problems of statistical inference (see, e.g., [8]). Recent work has sharpened this correspondence, showing that open sets in the weak topology correspond exactly to the statistical hypotheses that can be naturally deemed *verifiable* [14, 16]. We extend the correspondence to higher levels of the causal hierarchy, including the most refined and expansive "top" level consisting of all (well-founded) causal models. Lower levels and natural subspaces (e.g., corresponding to prominent causal assumption classes) emerge as coarsenings and continuous projections of this largest space. As an illustration of the general approach, we prove a topological version of the causal hierarchy theorem from [2]. Rather than showing that collapse happens only in a measure zero set as in [2], our Theorem 3 show that collapse is *topologically meager*. Conceptually, this highlights a different (but complementary) intuition: not only is collapse exceedingly unlikely in the sense of measure, meagerness implies that collapse could never be *statistically verified*. Correlatively, this implies that any causal assumption that would generally allow us to infer counterfactual probabilities from experimental (or "interventional") probabilities must itself be statistically unverifiable (Corollary 1).

To derive such a result we actually show something slightly stronger (see Lem. 2): even with respect to the subspace of models consistent with a fixed *temporal order* on variables, the causal hierarchy theorem holds. Merely knowing the temporal order of the variables is not enough to render collapse of the hierarchy a statistically verifiable proposition. Furthermore, we show that the *witness to collapse* can be taken as any of the well-known counterfactual "probabilities of causation" (see, e.g., [27]): probabilities of necessity, sufficiency, necessity *and* sufficiency, enablement, or disablement. That is, none of these important quantities are fully determined by experimental data except in a meager set.

In §2 we give background on causal models, and in §3 we present a model-theoretic characterization of the causal hierarchy as a sequence of spaces. Topology is introduced in §4, and the main results about collapse appear in §5. For the technical results, we include proof sketches in the main text to provide the core intuitions, relegating some of the details to an exhaustive technical appendix, which also includes additional supplementary material.

## 2 Structural Causal Models

A fundamental building block in the theory of causality is the *structural causal model* [26, 35, 28] or SCM, which formalizes the notion of a data-generating process. In addition to specifying data-generating distributions, these models also specify the generative mechanisms that produce them. For the purpose of causal inference and learning, SCMs provide a broad, fine-grained hypothesis space.

The notions in this section have their usual definition following, e.g., [28], but we have recast them in the standard language of Borel probability spaces so as to handle the case of infinitely many variables rigorously. We start with notation, basic assumptions, and some probability theory.

**Notation.** The signature (or range) of a variable $V$ is denoted $\chi_V$. Where $\mathbf{S}$ is a set of variables, let $\chi_{\mathbf{S}} = \bigtimes_{S \in \mathbf{S}} \chi_S$. Given an indexed family of sets $\{S_\beta\}_{\beta \in B}$ and elements $s_\beta \in S_\beta$, let $(s_\beta)_\beta$ denote the tuple whose element at index $\beta$ is $s_\beta$, for all $\beta$. For $B' \subset B$ write $\pi_{B'} : \bigtimes_{\beta \in B} S_\beta \to \bigtimes_{\beta \in B'} S_\beta$ for the *projection map* sending each $(s_\beta)_{\beta \in B} \mapsto (s_{\beta'})_{\beta' \in B'}$; abbreviate $\pi_{\beta'} = \pi_{\{\beta'\}}$, where $\beta' \in B$.

The reader is referred to standard texts [21, 5] for elaboration on the concepts used below.

**Definition 1** (Topology). For discrete spaces (like $\chi_S$, for a single categorical variable $S$) we use the discrete topology and for product spaces (like $\chi_{\mathbf{S}}$ for a set of variables $\mathbf{S}$) we use the product topology. Note that the so-called *cylinder sets* of the form $\pi_{\mathbf{Y}}^{-1}(\{\mathbf{y}\})$ for finite subsets $\mathbf{Y} \subset \mathbf{S}$ and $\mathbf{y} \in \chi_{\mathbf{Y}}$ form a basis for the product topology on $\chi_{\mathbf{S}}$. This cylinder set is a subset of $\chi_{\mathbf{S}}$, and contains exactly those valuations agreeing with the value $\pi_Y(\mathbf{y})$ specified in $\mathbf{y}$ for $Y$, for every $Y \in \mathbf{Y}$. Following standard statistical notation this cylinder is abbreviated as simply $\mathbf{y}$.

**Definition 2** (Probability). Where $\vartheta$ is a topological space write $\mathcal{B}(\vartheta)$ for its Borel $\sigma$-algebra of measurable subsets. Let $\mathfrak{P}(\vartheta)$ be the set of probability measures on $\mathcal{B}(\vartheta)$. Specifically, elements of $\mathfrak{P}(\vartheta)$ are functions $\mu : \mathcal{B}(\vartheta) \to [0,1]$ assigning a probability to each measurable set such that $\mu(\vartheta) = 1$ and $\mu\big(\bigcup_{i=1}^{\infty}(S_i)\big) = \sum_{i=1}^{\infty} \mu(S_i)$ for each sequence $S_1, S_2, \ldots$ of pairwise disjoint sets from $\mathcal{B}(\vartheta)$. A map $f : \vartheta_1 \to \vartheta_2$ is said to be *measurable* if $f^{-1}(S_2) \in \mathcal{B}(\vartheta_1)$ for every $S_2 \in \mathcal{B}(\vartheta_2)$.

**Fact 1** (Lemma 1.9.4 [5]). A Borel probability measure is determined by its values on a basis.

## 2.1 SCMs, Observational Distributions

Let $\mathbf{V}$ be a set of *endogenous variables*. We assume for simplicity every variable $V \in \mathbf{V}$ is dichotomous with $\chi_V = \{0,1\}$, although the results here generalize to any larger countable range. Influences among endogenous variables are the main phenomena our formalism aims to capture. A well-founded[1] *direct influence* relation $\to$ on $\mathbf{V}$ encapsulates the notion of one endogenous variable possibly influencing another. For each $V \in \mathbf{V}$, we call $\{V' \in \mathbf{V} : V' \to V\} = \mathbf{Pa}(V)$ the *parents* of $\mathbf{V}$. We assume every set $\mathbf{Pa}(V)$ is finite; this condition is called *local finiteness*. These two assumptions (well-foundedness and local finiteness) generalize the common *recursiveness* assumption to the infinitary setting, and have an alternative characterization in terms of "temporal" orderings:

**Fact 2.** Say that a total order $\prec$ on $\mathbf{V}$ is $\omega$-*like* if every node has finitely many predecessors: for each $V \in \mathbf{V}$, the set $\{V' : V' \prec V\}$ is finite. Then the influence relation $\to$ is extendible to an $\omega$-like order iff $\to$ is well-founded and locally finite.

In addition to endogenous variables, causal models have *exogenous variables* $\mathbf{U}$. Each endogenous $V$ depends on a subset $\mathbf{U}(V) \subset \mathbf{U}$ of "exogenous parents" and uncertainty enters via exogenous noise, that is, a distribution from $\mathfrak{P}(\chi_{\mathbf{U}})$. A *structural function* (or *mechanism*) for $V \in \mathbf{V}$ is a measurable $f_V : \chi_{\mathbf{Pa}(V)} \times \chi_{\mathbf{U}(V)} \to \chi_V$ mapping parental endogenous and exogenous valuations to values.

**Definition 3.** A *structural causal model* is a tuple $\mathcal{M} = \langle \mathbf{U}, \mathbf{V}, \{f_V\}_{V \in \mathbf{V}}, P \rangle$ where $\mathbf{U}$ is a collection of exogenous variables, $\mathbf{V}$ is a collection of endogenous variables, $f_V$ is a structural function for each $V \in \mathbf{V}$, and $P \in \mathfrak{P}(\chi_{\mathbf{U}})$ is a probability measure on (the Borel $\sigma$-algebra of) $\chi_{\mathbf{U}}$.

As is well known, recursiveness implies that each $\mathbf{u} \in \chi_{\mathbf{U}}$ induces a unique $\mathbf{v} \in \chi_{\mathbf{V}}$ that solves the simultaneous system of structural equations $\{V = f_V\}_V$:

**Proposition 1.** Any SCM $\mathcal{M}$ with well-founded, locally finite parent relation $\to$ induces a unique measurable $m^{\mathcal{M}} : \chi_{\mathbf{U}} \to \chi_{\mathbf{V}}$ such that $f_V\big(\pi_{\mathbf{Pa}(V)}(m^{\mathcal{M}}(\mathbf{u})), \pi_{\mathbf{U}(V)}(\mathbf{u})\big) = \pi_V\big(m^{\mathcal{M}}(\mathbf{u})\big)$ for all $\mathbf{u} \in \chi_{\mathbf{U}}$ and $V \in \mathbf{V}$.

Measurability then entails that the exogenous noise $P$ induces a distribution on joint valuations of $\mathbf{V}$, called the *observational distribution*, which characterizes passive observations of the system.

**Definition 4.** The observational distribution $p^{\mathcal{M}} \in \mathfrak{P}(\chi_{\mathbf{V}})$ is defined on open sets by $p^{\mathcal{M}}(\mathbf{y}) = P\big((m^{\mathcal{M}})^{-1}(\mathbf{y})\big)$. Here recall that $\mathbf{y}$ represents a cylinder subset (Definition 1) of $\chi_{\mathbf{V}}$.

---

[1]See Appendix A for additional background on orders and relations.

## 2.2 Interventions

What makes SCMs distinctively causal is the way they accommodate statements about possible manipulations of a causal setup capturing, e.g., observations resulting from a controlled experimental trial. This is formalized in the following definition.

**Definition 5.** An *intervention* is a choice of a finite subset of variables $\mathbf{W} \subset \mathbf{V}$ and $\mathbf{w} \in \chi_{\mathbf{W}}$. This intervention is written $\mathbf{W} := \mathbf{w}$, and we let $A$ be the set of all interventions. Under this intervention, each $W \in \mathbf{W}$ is held fixed to its value $\pi_W(\mathbf{w}) \in \chi_W$ in $\mathbf{w}$ while the mechanism for any $V \in \mathbf{V} \setminus \mathbf{W}$ is left unchanged. Specifically, where $\mathcal{M}$ is as in Definition 3, the manipulated model for $\mathbf{W} := \mathbf{w}$ is the model $\mathcal{M}_{\mathbf{W}:=\mathbf{w}} = \langle \mathbf{U}, \mathbf{V}, \{f_V^{\mathbf{W}:=\mathbf{w}}\}_{V \in \mathbf{V}}, P \rangle$ where

$$f_V^{\mathbf{W}:=\mathbf{w}} = \begin{cases} f_V, & V \notin \mathbf{W} \\ \text{constant func. mapping to } \pi_V(\mathbf{w}), & V \in \mathbf{W}. \end{cases}$$

The *interventional* or *experimental distribution* $p^{\mathcal{M}_{\mathbf{W}:=\mathbf{w}}} \in \mathfrak{P}(\chi_{\mathbf{V}})$ is just the observational distribution for the manipulated model $\mathcal{M}_{\mathbf{W}:=\mathbf{w}}$, and it encodes the probabilities for an experiment in which the variables $\mathbf{W}$ are fixed to the values $\mathbf{w}$.

**Remark 1.** Empty interventions $\varnothing := ()$ are just passive observations, i.e., $p^{\mathcal{M}_{\varnothing:=()}} = p^{\mathcal{M}}$.

## 2.3 Counterfactuals

By permitting multiple manipulated settings to share exogenous noise, not only the distribution arising from a single manipulation, but also joint distributions over multiple can be considered. These are often called *counterfactuals*. The set $\mathfrak{P}(\chi_{A \times \mathbf{V}})$ encompasses the combined joint distributions over $\mathbf{V}$ for any combination of interventions from $A$. A basis for the space $\chi_{A \times \mathbf{V}}$ are the cylinder sets of the following form, for some sequence $(\mathbf{X} := \mathbf{x}, \mathbf{Y}), \ldots, (\mathbf{W} := \mathbf{w}, \mathbf{Z})$ of pairs, where $\mathbf{Y}, \ldots, \mathbf{Z} \subset \mathbf{V}$ are finite, and $\mathbf{X} := \mathbf{x}, \ldots, \mathbf{W} := \mathbf{w} \in A$ are interventions:

$$\pi_{\{\mathbf{X}:=\mathbf{x}\} \times \mathbf{Y}}^{-1}(\{\mathbf{y}\}) \cap \cdots \cap \pi_{\{\mathbf{W}:=\mathbf{w}\} \times \mathbf{Z}}^{-1}(\{\mathbf{z}\}).$$

We will abbreviate this open set as $\mathbf{y}_{\mathbf{x}}, \ldots, \mathbf{z}_{\mathbf{w}}$, writing, e.g. simply $\mathbf{x}$ for the intervention $\mathbf{X} = \mathbf{x}$.

**Definition 6.** Given $\mathcal{M}$, define a counterfactual distribution $p_{\text{cf}}^{\mathcal{M}} \in \mathfrak{P}(\chi_{A \times \mathbf{V}})$ on a basis as follows:

$$p_{\text{cf}}^{\mathcal{M}}(\mathbf{y}_{\mathbf{x}}, \ldots, \mathbf{z}_{\mathbf{w}}) = P\big((m^{\mathcal{M}_{\mathbf{X}:=\mathbf{x}}})^{-1}(\mathbf{y}) \cap \cdots \cap (m^{\mathcal{M}_{\mathbf{W}:=\mathbf{w}}})^{-1}(\mathbf{z})\big).$$

Here, the letters $\mathbf{y}, \ldots, \mathbf{z}$ on the right-hand side abbreviate the respective cylinder sets (Definition 1) $\pi_{\mathbf{Y}}^{-1}(\{\mathbf{y}\}), \ldots, \pi_{\mathbf{Z}}^{-1}(\{\mathbf{z}\})$.

**Remark 2.** Marginalizing $p_{\text{cf}}^{\mathcal{M}}$ to any single intervention $\mathbf{W} := \mathbf{w}$ yields $p^{\mathcal{M}_{\mathbf{W}:=\mathbf{w}}}$. If $\chi_{\mathbf{U}}$ is finite, we obtain a familiar [13] sum formula $p_{\text{cf}}^{\mathcal{M}}(\mathbf{y}_{\mathbf{x}}, \ldots, \mathbf{z}_{\mathbf{w}}) = \sum_{\{\mathbf{u} \mid m^{\mathcal{M}_{\mathbf{X}:=\mathbf{x}}}(\mathbf{u}) \in \mathbf{y}, \ldots, m^{\mathcal{M}_{\mathbf{W}:=\mathbf{w}}}(\mathbf{u}) \in \mathbf{z}\}} P(\mathbf{u})$.

**Example 1.** As a very simple example (drawn from [28, 2]), just to illustrate the previous definitions and notation, consider a scenario with two binary exogenous variables $\mathbf{U} = \{U_1, U_2\}$ and two binary endogenous variables $\mathbf{V} = \{X, Y\}$. Let $U_1, U_2$ both be uniformly distributed, and define $f_X : \chi_{U_1} \to \chi_X$ to be the identity, and $f_Y : \chi_X \times \chi_{U_2} \to \chi_Y$ by $f_Y(u, x) = ux + (1-u)(1-x)$. This fully defines an SCM $\mathcal{M}$ with influence $X \to Y$, and produces an observational distribution $p^{\mathcal{M}}$ such that $p^{\mathcal{M}}(x, y) = 1/4$ for all four settings $X = x, Y = y$.

The space $A$ of interventions in this example includes the empty intervention and all combinations of $X := x$ and $Y := y$, with $x, y \in \{0, 1\}$. Notably, all interventional distributions here collapse to observational distributions, e.g., $p^{\mathcal{M}_{X:=x}}(X, Y) = p^{\mathcal{M}}(X, Y)$, for both values of $x$. Thus, "experimental" manipulations of this system reveal little interesting causal structure. The counterfactual distribution $p_{\text{cf}}^{\mathcal{M}}$, however, does not trivialize. For instance, $p_{\text{cf}}^{\mathcal{M}}((X = 1, Y = 1), (X := 0, Y = 0)) = 1/2$. This term is known as the *probability of necessity and sufficiency* [27], which we can abbreviate by $p_{\text{cf}}^{\mathcal{M}}(y_x, y'_{x'})$. Note that $p_{\text{cf}}^{\mathcal{M}}(y_x, y'_{x'}) \neq p_{\text{cf}}^{\mathcal{M}}(y_x) p_{\text{cf}}^{\mathcal{M}}(y'_{x'}) = 1/4$. Similarly, $p_{\text{cf}}^{\mathcal{M}}(y'_x, y_{x'}) = 1/2$.

## 2.4 SCM classes

We now define several subclasses of SCMs that we will use throughout the paper. Notably, we do not require their endogenous variable sets $\mathbf{V}$ to be finite. It is infinite in many applications, for instance, in time series models, or generative models defined by probabilistic programs (see, e.g., [18, 36]). Because the proofs call for slightly different methods, we deal with the infinite and finite cases separately. We make one additional assumption in the infinite case.

**Definition 7.** $\mu \in \mathfrak{P}(\vartheta)$ is *atomless* if $\mu(\{t\}) = 0$ for each $t \in \vartheta$; $\mathcal{M}$ is atomless if $p_{\mathrm{cf}}^{\mathcal{M}}$ is atomless.

Intuitively, an atomless distribution is one in which weight is always "smeared" out continuously and there are no point masses; infinitely many fair coin flips, for example, generate an atomless distribution as the probability of obtaining any given infinite sequence is zero.

**Definition 8.** For the remainder of the paper, fix a countable endogenous variable set $\mathbf{V}$. Define the following classes of SCMs:

$$\mathfrak{M}_{\prec} = \text{SCMs over } \mathbf{V} \text{ whose influence relation is extendible to the } \omega\text{-like order } \prec;$$

$$\mathfrak{M}_X = \text{SCMs over } \mathbf{V} \text{ in which the variable } X \text{ has no parents: } \mathbf{Pa}(X) = \varnothing;$$

$$\mathfrak{M} = \text{all SCMs over } \mathbf{V} = \bigcup_{\prec} \mathfrak{M}_{\prec} = \bigcup_X \mathfrak{M}_X.$$

If $\mathbf{V}$ is infinite then all SCMs in the classes above are assumed to be atomless.

# 3 The Causal Hierarchy

Implicit in §2, and indeed in much of the literature on causal inference, is a hierarchy of causal expressivity. Following the metaphor offered in [29], it is natural to characterize three levels of the hierarchy as the *observational*, *interventional* (experimental), and *counterfactual* (explanatory). Drawing on recent work [2, 19] we make this characterization explicit. The levels will be defined in descending order of causal expressivity (the reverse of §2). Fig. 1(a) summarizes our definitions.

Higher levels determine lower levels—counterfactuals determine interventionals, and the observational is just an (empty) interventional. Thus movement "downward" in the causal hierarchy corresponds to a kind of projection. For indexed $\{S_\beta\}_{\beta \in B}$ and $B' \subset B$ let $\varsigma_{B'} : \mathfrak{P}(\bigtimes_{\beta \in B} S_\beta) \to \mathfrak{P}(\bigtimes_{\beta \in B'} S_\beta)$ be the *marginalization* map taking a joint distribution to its marginal on $B'$.

**Definition 9.** Define three composable *causal projections* $\{\varpi_i\}_{1 \leq i \leq 3}$ with signatures and definitions

$$\varpi_3 : \mathfrak{M} \to \mathfrak{P}(\chi_{A \times \mathbf{V}}), \quad \varpi_2 : \mathfrak{P}(\chi_{A \times \mathbf{V}}) \to \bigtimes_{\alpha \in A} \mathfrak{P}(\chi_{\mathbf{V}}), \quad \varpi_1 : \bigtimes_{\alpha \in A} \mathfrak{P}(\chi_{\mathbf{V}}) \to \mathfrak{P}(\chi_{\mathbf{V}});$$

$$\varpi_3 : \mathcal{M} \mapsto p_{\mathrm{cf}}^{\mathcal{M}}, \quad \varpi_2 : \mu_3 \mapsto \left( \varsigma_{\{\alpha\} \times \mathbf{V}}(\mu_3) \right)_{\alpha \in A}, \quad \varpi_1 : (\mu_\alpha)_{\alpha \in A} \mapsto \mu_{\varnothing := ()} = \pi_{\varnothing := ()} \left( (\mu_\alpha)_\alpha \right).$$

The *causal hierarchy* consists of three sets $\{\mathfrak{S}_i\}_{1 \leq i \leq 3}$ defined as images or projections of $\mathfrak{M}$:

$$\mathfrak{S}_3 = \varpi_3(\mathfrak{M}), \quad \mathfrak{S}_2 = \varpi_2(\mathfrak{S}_3), \quad \mathfrak{S}_1 = \varpi_1(\mathfrak{S}_2).$$

These are the three *Levels* of the hierarchy. The definitions cohere with those of §2 (and, e.g., [28, 2]):

**Fact 3.** Let $\mathcal{M} \in \mathfrak{M}$. Then $\mu_3 = \varpi_3(\mathcal{M}) \in \mathfrak{S}_3$ trivially coincides with its counterfactual distribution as defined in §2.3, while $(\mu_\alpha)_\alpha = \varpi_2(\mu_3) \in \mathfrak{S}_2$ coincides with the indexed family of all its interventional distributions (§2.2), i.e., $\pi_{\mathbf{W} := \mathbf{w}} \left( (\mu_\alpha)_\alpha \right) = p^{\mathcal{M}_{\mathbf{W} := \mathbf{w}}}$ for each $\mathbf{W} := \mathbf{w} \in A$. Finally $\mu = \varpi_1 \left( (\mu_\alpha)_\alpha \right) \in \mathfrak{S}_1$ coincides with its observational distribution (§2.1).

Thus, e.g., $\mathfrak{S}_3$ is the set of counterfactual distributions that are consistent with at least some SCM from $\mathfrak{M}$. It is a fact that $\mathfrak{S}_3 \subsetneq \mathfrak{P}(\chi_{A \times \mathbf{V}})$ and similarly not every interventional family belongs to $\mathfrak{S}_2$; see Appendix B for explicit characterizations. At the observational level, this is simple:

**Fact 4.** $\mathfrak{S}_1 = \mathfrak{P}(\chi_{\mathbf{V}})$ in the finite case. In the infinite case, $\mathfrak{S}_1 = \{\mu \in \mathfrak{P}(\chi_{\mathbf{V}}) : \mu \text{ is atomless}\}$.

We will also use the subsets $\{\mathfrak{S}_i^{\prec}\}_i$ and $\{\mathfrak{S}_i^X\}_i$, which are defined analogously but via projection from $\mathfrak{M}_{\prec}$ and $\mathfrak{M}_X$ respectively.

## 3.1 Problems of Causal Inference

As elucidated in [29, 2], the causal hierarchy helps characterize many standard problems of causal inference, in as far as these problems typically involve ascending levels of the hierarchy. Some examples include:

1. Classical identifiability: given observational data about some variables in $\mathbf{V}$, estimate a *causal effect* of setting variables $\mathbf{X}$ to values $\mathbf{x}$ [26, 35]. In the notation here, given information about $p^{\mathcal{M}}(\mathbf{V})$, can we determine $p^{\mathcal{M}_{\mathbf{X} := \mathbf{x}}}(\mathbf{Y})$?

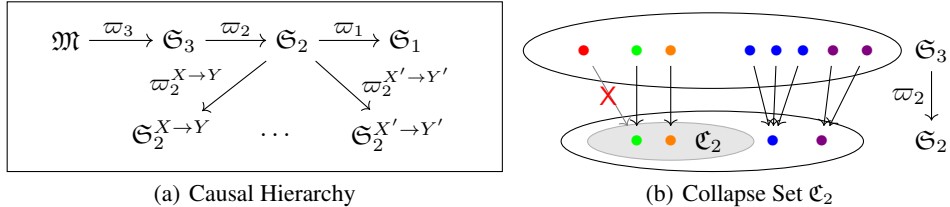

|  |  |
|---|---|
| (a) Causal Hierarchy | (b) Collapse Set $\mathfrak{C}_2$ |

Figure 1: (a) $\mathfrak{S}_3$ can be seen as a coarsening of $\mathfrak{M}$, abstracting from irrelevant "intensional" details. $\mathfrak{S}_2$ is obtained from $\mathfrak{S}_3$ by marginalization (also a coarsening), while $\mathfrak{S}_1$ is a projection of $\mathfrak{S}_2$ via the "empty" intervention. Each map $\varpi_i$, $i = 1, 2, 3$, is continuous for the respective weak topologies (Prop. 4). The projections $\varpi_2^{X \to Y}$ from $\mathfrak{S}_2$ to the 2VE-spaces are likewise continuous (Prop. 4). (b) The shaded region, $\mathfrak{C}_2 \subset \mathfrak{S}_2$, is the collapse set in which Level 2 facts determine all Level 3 facts: those points in $\mathfrak{S}_2$ whose $\varpi_2$-preimage in $\mathfrak{S}_3$ is a singleton set. The main result of this paper is that $\mathfrak{C}_2$ is *meager* in weak topology on $\mathfrak{S}_2$ (Thm. 3). This means $\mathfrak{C}_2$ contains no open subset, which by Thm. 2 implies no part of $\mathfrak{C}_2$ is statistically verifiable, even with infinitely many ideal experiments.

2. General identifiability: given a mix of observational data and limited experimental data— that is, information about $p^{\mathcal{M}}(\mathbf{V})$ as well as some experimental distributions of the form $p^{\mathcal{M}_{\mathbf{w}:=\mathbf{w}}}(\mathbf{V})$—determine $p^{\mathcal{M}_{\mathbf{x}:=\mathbf{x}}}(\mathbf{Y})$ [38, 22].

3. Structure learning: given observational data, and perhaps experimental data, infer properties of the underlying causal influence relation $\to$ [35, 30].

4. Counterfactual estimation: given a combination of observational and experimental data, infer a counterfactual quantity, such as probability of necessity [31], or probability of necessity and sufficiency [27, 37] (see also §3.3 below).

5. Global identifiability: given observational data drawn from $p^{\mathcal{M}}(\mathbf{V})$ infer the full counterfactual distribution $p_{\text{cf}}^{\mathcal{M}}(A \times \mathbf{V})$ [34, 10].

This is not an exhaustive list, and these problems are not all independent of one another. They are also all unsolvable in general. Problems 1, 2, and 3 involve ascending to Level 2 given information at Level 1 (and perhaps partial information at Level 2); problems 4 and 5 ask us to ascend to Level 3 given only Level 1 (and perhaps also Level 2) information. The upshot of the causal hierarchy theorem from [2] is that these steps are impossible without assumptions, formalizing the common wisdom, "no causes in, no causes out" [7]. To understand the statement of the causal hierarchy theorem—and our topological version of it—we explain what it means for the hierarchy to collapse.

### 3.2 Collapse of the Hierarchy

In the present setting a *collapse* of the hierarchy can be understood in terms of injectivity of the functions $\varpi_i$. For $i = 1, 2$ let $\mathfrak{C}_i \subset \mathfrak{S}_i$ be the injective fibers of $\varpi_i$, i.e., $\mathfrak{C}_i = \{\mu_i \in \mathfrak{S}_i : \mu_{i+1} = \mu'_{i+1} \text{ whenever } \varpi_i(\mu_{i+1}) = \varpi_i(\mu'_{i+1}) = \mu_i\}$. Every element $\mu \in \mathfrak{C}_i$ is a witness to (global) collapse of the hierarchy: knowing $\mu$ would be sufficient to determine the Level $i + 1$ facts completely.

A first observation is that $\varpi_1$ is *never* injective. In other words, the distribution $p^{\mathcal{M}}(\mathbf{V})$ never determines all the interventional distributions $p^{\mathcal{M}_{\mathbf{x}:=\mathbf{x}}}(\mathbf{Y})$. This is essentially a way of stating that correlation never implies causation absent assumptions. (See also [2, Thm. 1].)

**Proposition 2.** $\mathfrak{C}_1 = \varnothing$. That is, Level 2 *never* collapses to Level 1 without assumptions.

To overcome this formidable inferential barrier, researchers often assume we are not working in the "full" space $\mathfrak{M}$ of all causal models, but rather some proper subset embodying a range of causal assumptions. This may effectively eliminate counterexamples to collapse (cf. Fig. 1(b)). For problems of type 1 or 2 (from the list above in §3.1) it is common to assume we are only dealing with models whose graph (direct influence relation) $\to$ satisfies a fixed set of properties. For problems of type 3 it is common to assume that $p^{\mathcal{M}}$ and $\to$ relate in some way (for instance, through an assumption like *faithfulness* or *minimality* [35]). All of these problems become solvable with sufficiently strong assumptions about the form of the functions $\{f_V\}_V$ or the probability space $P$.

In some cases, the relevant causal assumptions are justified by appeal to background or expert knowledge. In other cases, however, an assumption will be justified by the fact that it rules out only

a "small" or "negligible" or "measure zero" part of the full set $\mathfrak{M}$ of possibilities. As emphasized by a number of authors [12, 39, 23], not all "small" subsets are the same, and it seems reasonable to demand further justification for eliminating one over another. We believe that the framework presented here can contribute to this positive project, but our immediate interest is in solidifying and clarifying limitative results about what cannot be done.

The issue of collapse becomes especially delicate when we turn to $\mathfrak{C}_2$. When do interventional distributions fully determine counterfactual distributions? In contrast to Prop. 2 we have:

**Proposition 3.** $\mathfrak{C}_2 \neq \varnothing$. That is, there exists an SCM in which Level 3 collapses to Level 2.

*Proof sketch.* As a very simple example in the finite case, any fully deterministic SCM will result in collapse. This is because, if $(\mu_\alpha)_{\alpha \in A}$ are all binary-valued then the measure $\mu_3 \in \mathfrak{P}\big( \bigtimes_{\alpha \in A} \chi_\mathbf{V} \big)$ that produces the marginals $\mu_\alpha$ is completely determined: each $\mu_\alpha$ specifies an element of $\chi_\mathbf{V}$, so $\mu_3$ must assign unit probability to the tuple that matches $\mu_\alpha$ at the $\alpha$ projection. In the infinite case, any example must be non-deterministic by atomlessness, but collapse is still possible; see Example 2 in Appendix B. $\qquad\square$

### 3.3 Probabilities of Causation

A handful of counterfactual quantitites over two given variables, collected below, have been particularly prominent in the literature (e.g., [27]). Our main result will show that *any* of these six quantities (for any two fixed variables) is robust against collapse. Below, fix two distinct variables $Y \neq X \in \mathbf{V}$ and distinct values $x \neq x' \in \chi_X, y \neq y' \in \chi_Y$.

**Definition 10.** The *probabilities of causation* are the following quantities:

$$P(y_x, y'_{x'}) : \text{probability of necessity and sufficiency}$$
$$P(y'_x, y_{x'}) : \text{converse prob. of necessity and sufficiency}$$
$$P(y'_{x'} \mid x, y) : \text{prob. of necessity} \qquad P(y_x \mid x', y') : \text{ prob. of sufficiency}$$
$$P(y'_{x'} \mid y) : \text{prob. of disablement} \qquad P(y_x \mid y') : \text{ prob. of enablement}$$

Consider, for example, the probability of necessity and sufficiency (PNS), which is the joint probability that $Y$ would take on value $y$ if $X$ is set by intervention to $x$, and $y'$ if $X$ is set to $x'$. PNS has been thoroughly studied [27, 37, 1], in part due to its widespread relevance: from medical treatment to online advertising, we would like to assess which interventions are likely to be both *necessary* and *sufficient* for a given outcome. Using the notation from §2.3, PNS concerns the measure of sets $y_x, y'_{x'} = \pi^{-1}_{(X:=x,Y)}(\{y\}) \cap \pi^{-1}_{(X:=x',Y)}(\{y'\})$.

The probabilities of causation are paradigmatically Level 3, and we will be interested in their manifestations at Level 2. In that direction we introduce a small part of $\mathfrak{S}_2$, just enough to witness the behavior of $Y$ (and $X$) under the empty intervention and the two possible interventions on $X$:

**Definition 11.** Let $A_X = \{\varnothing := (), X := 0, X := 1\}$. Define a small subspace $\mathfrak{S}_2^{X \to Y} \subset \bigtimes_{\alpha \in A_X} \mathfrak{P}(\chi_{\{X,Y\}})$ as the image of the map $\varpi_2^{X \to Y} = \big(\varsigma_{\{X,Y\}} \times \varsigma_{\{X,Y\}} \times \varsigma_{\{X,Y\}}\big) \circ \pi_{A_X}$ (see Fig. 1(a)). Call $\mathfrak{S}_2^{X \to Y}$ a *two-variable effect* (2VE) *space*; fixing $X$, we have a 2VE-space for each $Y$.

It is known in the literature that the probabilities of causation are not identifiable from the data $p(X,Y)$, $p(Y_x)$, and $p(Y_{x'})$ (see, e.g., [1] for PNS). As part of our proof of Theorem 3 below, we will strengthen this considerably to show them all to be *generically* unidentifiable, in a topological sense to be made precise.

## 4 The Weak Topology

We now demonstrate how $\mathfrak{S}_1, \mathfrak{S}_2$ and $\mathfrak{S}_3$ can be topologized. In general, given a space $\vartheta$ and the set $\mathfrak{S} = \mathfrak{P}(\vartheta)$ of Borel probability measures on $\vartheta$, a natural topology on $\mathfrak{S}$ can be defined as follows:

**Definition 12.** For a sequence $(\mu_n)_n$ of measures in $\mathfrak{S}$, write $(\mu_n)_n \Rightarrow \mu$ and say it *converges weakly* [4, p. 7] to $\mu$ if $\int_\vartheta f \, d\mu_n \to \int_\vartheta f \, d\mu$ for all bounded, continuous $f : \vartheta \to \mathbb{R}$. Then the *weak topology* $\tau^\mathrm{w}$ on $\mathfrak{S}$ is that with the following closed sets: $E \subset \mathfrak{S}$ is closed in $\tau^\mathrm{w}$ iff for any weakly convergent sequence $(\mu_n)_n \Rightarrow \mu$ in which every $\mu_n \in E$, the limit point $\mu$ is in $E$.

There are several alternative characterizations of $\tau^{\mathrm{w}}$, which hold under very general conditions. For instance, it coincides with the topology induced by the so called Lévy-Prohorov metric [4]. The most useful characterization for our purposes is that it can be generated by subbasic open sets of the form

$$\{\mu : \mu(X) > r\} \tag{1}$$

with $X$ ranging over basic clopens in $\vartheta$ and $r$ over rationals (see, e.g., [16, Lemma A.5]).

Conceptually, the explication of $\tau^{\mathrm{w}}$ in terms of weak convergence strongly suggests a connection with statistical learning. We now make this connection precise, building on existing work [8, 14, 16].

## 4.1 Connection to Learning Theory

Roughly speaking, we will say a hypothesis $H \subseteq \mathfrak{S}$ is *statistically verifiable* if there is some error bound $\epsilon$ and a sequence of statistical tests that converge on $H$ with error at most $\epsilon$, when data are generated from $H$. More formally, a *test* is a function $\lambda : \vartheta^n \to \{\text{accept}, \text{reject}\}$, where $\vartheta^n$ is the $n$-fold product of $\vartheta$, viz. finite data streams from $\vartheta$. The interest is in whether a "null" hypothesis can be rejected given data observed thus far. The *boundary* of a set $A \subseteq \vartheta$, written $\mathrm{bd}(A)$, is the difference of its closure and its interior. Intuitively, a learner will not be able to decide whether to accept or reject on the boundary. Consequently it is assumed that $\lambda$ is *feasible* in the sense that the boundary of its acceptance zone (in the product topology on $\vartheta^n$) always has measure 0, i.e., $\mu^n[\mathrm{bd}(\lambda^{-1}(\text{accept}))] = 0$ for every $\mu \in \mathfrak{S}$, where $\mu^n$ is the $n$-fold product measure of $\mu$.

Say a hypothesis $H \subseteq \mathfrak{S}$ is *verifiable* [14] if there is $\epsilon > 0$ and a sequence $(\lambda_n)_{n \in \mathbb{N}}$ of feasible tests (of the complement of $H$ in $\mathfrak{S}$, i.e., the "null hypothesis") such that

1. $\mu^n[\lambda_n^{-1}(\text{reject})] \leq \epsilon$ for all $n$, whenever $\mu \notin H$;
2. $\lim_{n \to \infty} \mu^n[\lambda_n^{-1}(\text{reject})] = 1$, whenever $\mu \in H$.

That is, to be verifiable we only require a sequence of tests that converges in probability to the true hypothesis in the limit of infinite data (requirement 2), while incurring (type 1) error only up to a given bound at finite stages (requirement 1). As an illustrative example, *conditional dependence* is verifiable [14]. This is a relatively lax notion of verifiability. For instance, the hypothesis need not also be *refutable* (and thus "decidable"). For our purposes this generality is a virtue: we want to show that certain hypotheses are not statistically verifiable by any method, even in this wide sense. The fundamental link between verifiability and the weak topology is the following, due to [14, 16]:

**Theorem 1.** A set $H \subseteq \mathfrak{S}$ is verifiable if and only if it is open in the weak topology.

## 4.2 Topologizing Causal Models

We now reinterpret $\tau^{\mathrm{w}}$ at each level of the causal hierarchy:

**Definition 13.** The *weak causal topology* $\tau_i^{\mathrm{w}}$, $1 \leq i \leq 3$, is the subspace topology on $\mathfrak{S}_i$, induced by

$$\text{if } i = 3 : \tau^{\mathrm{w}} \text{ on } \mathfrak{P}(\chi_{A \times \mathbf{V}}); \quad \text{if } i = 2 : \text{product of } \tau^{\mathrm{w}} \text{ on } \bigtimes_{\alpha \in A} \mathfrak{P}(\chi_{\mathbf{V}}); \quad \text{if } i = 1 : \tau^{\mathrm{w}} \text{ on } \mathfrak{P}(\chi_{\mathbf{V}}).$$

**Proposition 4.** All projections $\{\varpi_i\}_i, \varpi_2^{X \to Y}$ are continuous in the weak causal topologies.

A significant observation is that the learning theoretic interpretation, originally intended for $\tau_1^{\mathrm{w}}$, naturally extends to $\tau_2^{\mathrm{w}}$. While data streams at Level 1 amount to passive observations of $\mathbf{V}$, data streams at Level 2 can be seen as sequences of experimental results, i.e., observations of "potential outcomes" $\mathbf{Y}_{\mathbf{x}}$. To make verifiability as easy as possible we assume a learner can observe a sample from all conceivable experiments at each step. A learner is thus a function $\lambda : \mathcal{E}^n \to \{\text{accept}, \text{reject}\}$, where $\mathcal{E}^n = ((\chi_{\mathbf{V}})^n)_\alpha$ is the set of potential experimental observations over $n$ trials (with $\alpha$ indexing the experiments). Construing $\mathcal{E}^n$ as a product space we can again speak of *feasibility* of $\lambda$.

Recall that elements of $\mathfrak{S}_2$ are tuples $(\mu_\alpha)_{\alpha \in A}$ of measures. Say a hypothesis $H \subseteq \mathfrak{S}_2$ is *experimentally verifiable* if there is $\epsilon > 0$ and a sequence $(\lambda_n)_{n \in \mathbb{N}}$ of feasible tests such that 1 and 2 above hold, replacing $\mu^n[\lambda_n^{-1}(\text{reject})]$ with $\prod_\alpha \mu_\alpha^n[(\lambda_n^{-1}(\text{reject}))_\alpha]$. That is, when experimental data are drawn from the interventional distributions $(\mu_\alpha)_{\alpha \in A} \in H$, we require that the learner eventually converge on $H$ with bounded error at finite stages. We can then show (see Appendix C):

**Theorem 2.** A set $H \subseteq \mathfrak{S}_2$ is experimentally verifiable if and only if it is open in $\tau_2^{\mathrm{w}}$.

A similar result can be given for $(\mathfrak{S}_3, \tau_3^{\mathrm{w}})$, although it is less clear what the empirical content of this result would be. Note also that $\tau_1^{\mathrm{w}}, \tau_2^{\mathrm{w}}, \tau_3^{\mathrm{w}}$ give a sequence of increasingly fine topologies on the set of actual SCMs $\mathfrak{M}$ by simply pulling back the projections. The point is that $\tau_2^{\mathrm{w}}$ is the finest that has clear empirical significance, while $\tau_3^{\mathrm{w}}$ is the finest in terms of relevance to the causal hierarchy.

## 5 Collapse is Meager

Recall that a set $X \subseteq \vartheta$ is *nowhere dense* if every open set contains an open $Y$ with $X \cap Y = \varnothing$. A countable union of nowhere dense sets is said to be *meager* (or *of first category*). The complement of a meager set is *comeager*. Intuitively, a meager set is one that can be "approximated" by sets "perforated with holes" [25]. Meagerness is notably preserved under continuous preimages.

As discussed above, one intuition highlighted by the weak topology $\tau^{\mathrm{w}}$ is that open sets are the kinds of probabilistic propositions that could, in the limit of infinite data, be verified (Thms. 1, 2). Correlatively, meager sets in $\tau^{\mathrm{w}}$ are so negligible as to be unverifiable: as a meager set contains no non-empty open subsets (by the Baire Category Theorem [25]), it is statistically unverifiable. We will now show that the injective collapse set $\mathfrak{C}_2$ from §3.2 is topologically meager.

The crux is to identify a "good" comeager 2VE-subspace where collapse *never* occurs (with separation witnessed by probabilities of causation). In this subspace, the constraints circumscribing Level 3 have sufficient slack to make a tweak without thereby disturbing Level 2 (cf. Figure 2). We define the good set as the locus of a set of strict inequalities:

**Definition 14.** A family $(\mu_\alpha)_{\alpha \in A_X} \in \mathfrak{S}_2^{X \to Y}$ is $Y$-*good* if we have the following, abbreviating the members of $A_X$ as $(), x, x'$:

$$0 < \mu_x(y') - \mu_{()}(x, y') < \mu_{()}(x'), \tag{2}$$

$$0 < \mu_{()}(x', y') < \mu_{()}(x'). \tag{3}$$

**Lemma 1.** The subspace of $Y$-good families is comeager in $\mathfrak{S}_2^{X \to Y}$.

*Proof sketch.* The non-strict versions of (2), (3) hold universally, so the complement of the good set is defined by equalities. This is closed and contains no nonempty open by the weak subbasis (1). $\square$

Figure 2 presents the construction in a small, two-variable case, and Lemma 2 below is proven by generalizing it to arbitrary $\mathbf{V}$. Guaranteeing agreement on every interventional distribution in the general case is subtle (Appendix D): it has been observed that enlarging $\mathbf{V}$ can enable additional inferences (e.g., [32]), though the next result reflects a dependence on further assumptions.

**Lemma 2.** Suppose $\prec$ is an order in which $X$ comes first and $(\mu_\alpha)_{\alpha \in A} \in \mathfrak{S}_2^{\prec}$ is such that $\varpi_2^{X \to Y}\big((\mu_\alpha)_\alpha\big)$ is $Y$-good, and let $\varphi$ be PNS, the converse PNS, the probability of sufficiency, or the probability of enablement (Definition 10). Then for any $\mu_3 \in \mathfrak{S}_3^{\prec}$ such that $\varpi_2(\mu_3) = (\mu_\alpha)_\alpha$, there exists a $\mu_3' \in \mathfrak{S}_3^{\prec}$ such that $\mu_3$ and $\mu_3'$ disagree on $\varphi$.

Note that by reversing the roles of $x$ and $x'$, we may obtain the same for the probability of necessity and probability of disablement. The main theorem and its important learning-theoretic corollary are now straightforward.

**Theorem 3** (Topological Hierarchy). The set $\mathfrak{C}_2$ of points where all Level 3 facts are identifiable from Level 2 is meager in $(\mathfrak{S}_2, \tau_2^{\mathrm{w}})$. The preimage $\varpi_2^{-1}(\mathfrak{C}_2) = \mathfrak{C}_3$ is likewise meager in $(\mathfrak{S}_3, \tau_3^{\mathrm{w}})$.

*Proof.* Let $\mathfrak{D}_2^{X,Y} \subset \mathfrak{S}_2^X$ be the preimage under $\varpi_2^{X \to Y}$ of the set of $Y$-good tuples in $\mathfrak{S}_2^{X \to Y}$. Lemma 2 implies that $\mathfrak{C}_2 \cap \mathfrak{S}_2^X$ is contained in $\mathfrak{S}_2^X \setminus \mathfrak{D}_2^{X,Y}$, for *any* $Y \neq X$. Meanwhile, since $\varpi_2^{X \to Y}$ is continuous, Lemma 1 implies that $\mathfrak{S}_2^X \setminus \mathfrak{D}_2^{X,Y}$ is meager in $\mathfrak{S}_2^X$, and thereby also in $\mathfrak{S}_2$. Thus $\mathfrak{C}_2 = \bigcup_{X \in \mathbf{V}} \mathfrak{C}_2 \cap \mathfrak{S}_2^X$ is a countable union of meager sets, and hence meager. $\square$

**Corollary 1.** No causal hypothesis licensing arbitrary counterfactual inferences (and specifically those of the probabilities of causation) from observational and experimental data is itself statistically (even experimentally) verifiable.

| | $\mathcal{M}$ | | | |
|---|---|---|---|---|
| $u$ | $P(u)$ | $X_u$ | $Y_{x,u}$ | $Y_{x',u}$ |
| $u_0$ | $1/2$ | $x'$ | $y$ | $y$ |
| $u_1$ | $1/2$ | $x'$ | $y'$ | $y'$ |

(a) $Y$-good Model

| | $\mathcal{M}'$ | | | |
|---|---|---|---|---|
| $u$ | $P(u)$ | $X_u$ | $Y_{x,u}$ | $Y_{x',u}$ |
| $u_0$ | $1/2 - \varepsilon$ | $x'$ | $y$ | $y$ |
| $u_1$ | $\varepsilon$ | $x'$ | $y$ | $y'$ |
| $u_2$ | $1/2 - \varepsilon$ | $x'$ | $y'$ | $y'$ |
| $u_3$ | $\varepsilon$ | $x'$ | $y'$ | $y$ |

(b) Example Separating Levels 2 and 3

Figure 2: (a): the structural functions and exogenous noise for a model $\mathcal{M}$ with direct influence $X \to Y$. This $\mathcal{M}$ meets (2) and (3), so we may apply Lemma 2, constructing the model $\mathcal{M}'$ in (b), where $0 < \varepsilon < 1/2$. Note that $p_{\text{cf}}^{\mathcal{M}}(y_x, y'_{x'}) = 0$ while $p_{\text{cf}}^{\mathcal{M}'}(y_x, y'_{x'}) = \varepsilon$, so that the two models disagree on a Level 3 PNS quantity; on the other hand, it is easy to check agreement on all of Level 2. Similarly, $\mathcal{M}$ and $\mathcal{M}'$ disagree on the converse PNS, probability of sufficiency, and probability of enablement (Definition 10).

# 6    Conclusion

We introduced a general framework for topologizing spaces of causal models, including the space of all (discrete, well-founded) causal models. As an illustration of the framework we characterized levels of the causal hierarchy topologically, and proved a topological version of the causal hierarchy theorem from [2]. While the latter shows that collapse of the hierarchy (specifically of Level 3 to Level 2) is *exceedingly unlikely* in the sense of (Lebesgue) measure, we offer a complementary result: any condition guaranteeing that we could infer arbitrary Level 3 information from purely Level 2 information must be *statistically unverifiable*, even by experimental means. Both results capture an important sense in which collapse is "negligible" in the space of all possible models. As an added benefit, the topological approach extends seamlessly to the setting of infinitely many variables.

There are many natural extensions of these results. For instance, we have begun work on a version for continuous endogenous variables. Also of interest are subspaces embodying familiar causal assumptions or other well-studied coarsenings of SCMs (see, e.g., [23] on Bayesian networks, or [17, 15] on linear non-Gaussian models), which often render important inference problems solvable, though sometimes only "generically" so. In the opposite direction, we expect analogous hierarchy theorems to hold for extensions of the SCM concept, e.g., that dropping the well-foundedness or recursiveness requirements [6]. As emphasized by [2], a causal hierarchy theorem should not be construed as a purely limitative result, but rather as further motivation for understanding the whole range of causal-inductive assumptions, how they relate, and what they afford. We submit that the topological constructions presented here can help clarify and systematize this broader landscape.

### Acknowledgments

This material is based upon work supported by the National Science Foundation Graduate Research Fellowship Program under Grant No. DGE-16565. We are very grateful to the five anonymous NeurIPS reviewers for insightful and detailed comments and questions that led to significant improvements in the paper. We would also like to thank Jimmy Koppel, Krzysztof Mierzewski, Francesca Zaffora Blando, and especially Kasey Genin for helpful feedback on earlier versions.

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
