# A Topological Perspective on Causal Inference: Supplement

In this supplement we give proofs of all the main results in the text.

## A  Structural Causal Models (§2)

### A.1  Background on Relations and Orders

**Definition A.1.1.** Let $C$ be a set. Then a subset $R \subset C \times C$ is called a *binary relation* on $C$. We write $cRc'$ if $(c, c') \in R$. The binary relation $R$ is *well-founded* if every nonempty subset $D \subset C$ has a minimal element with respect to $R$, i.e., if for every nonempty $D \subset C$, there is some $d \in D$, such that there is no $d' \in D$ such that $d'Rd$. The binary relation $\prec \subset C \times C$ is a (strict) *total order* if it is irreflexive, transitive, and *connected*: either $c \prec c'$ or $c' \prec c$ for all $c \neq c' \in C$.

**Example 1.** The edges of a dag form a well-founded binary relation on its nodes. If $\mathbf{V} = \{V_n\}_{n \geq 0}$, then the binary relation $\to$ defined by $V_m \to V_n$ iff either $0 < m < n$ or $n = 0 < m$ is well-founded but not extendible to an $\omega$-like total order (see Fact 2) and not locally finite: $V_0$ has infinitely many predecessors $V_1, V_2, \ldots$

### A.2  Proofs

*Proof of Proposition 1.* We assume without loss that $\mathbf{U}(V) = \mathbf{U}$ for every $V \in \mathbf{V}$. For each $\mathbf{u} \in \chi_{\mathbf{U}}$, well-founded induction along $\to$ shows unique existence of a $m^{\mathcal{M}}(\mathbf{u}) \in \chi_{\mathbf{V}}$ solving $f_V\big(\pi_{\mathbf{Pa}(V)}(m^{\mathcal{M}}(\mathbf{u})), \mathbf{u}\big) = \pi_V(m^{\mathcal{M}}(\mathbf{u}))$ for each $V$. We claim the resulting function $m^{\mathcal{M}}$ is measurable. One has a clopen basis of cylinders, so it suffices to show each preimage $(m^{\mathcal{M}})^{-1}(v)$ is measurable. Recall that here $v$ denotes the cylinder set $\pi_V^{-1}(\{v\}) \in \mathcal{B}(\chi_{\mathbf{V}})$, for $v \in \chi_V$. Once again this can be established inductively. Note that

$$(m^{\mathcal{M}})^{-1}(v) = \bigcup_{\mathbf{p} \in \chi_{\mathbf{Pa}(V)}} \Big[ (m^{\mathcal{M}})^{-1}(\mathbf{p}) \cap \pi_{\mathbf{U}}\big(f_V^{-1}(\{v\}) \cap (\{\mathbf{p}\} \times \chi_{\mathbf{U}})\big) \Big].$$

which is a finite union (by local finiteness) of measurable sets (by the inductive hypothesis) and therefore measurable. Thus for any $\mathcal{M}$ the pushforward $p^{\mathcal{M}} = m_*^{\mathcal{M}}(P)$ is a measure on $\mathcal{B}(\chi_{\mathbf{V}})$ and gives the observational distribution (Definition 4). □

*Remark on Definition 6.* To see that $p_{\mathrm{cf}}^{\mathcal{M}}$ thus defined is a measure, note that $p_{\mathrm{cf}}^{\mathcal{M}} = p^{\mathcal{M}_A}$ and apply Proposition 1, where the model $\mathcal{M}_A$ is defined in Definition A.2.1. This is similar in spirit to the construction of "twinned networks" [2] or "single-world intervention graphs" [8]. □

**Definition A.2.1.** Given $\mathcal{M}$ as in Def. 3 and a collection of interventions $A$ form the following *counterfactual model* $\mathcal{M}_A = \langle \mathbf{U}, A \times \mathbf{V}, \{f_{(\alpha, V)}\}_{(\alpha, V)}, P \rangle$, over endogenous variables $A \times \mathbf{V}$. The counterfactual model has the influence relation $\to'$, defined as follows. Where $\alpha', \alpha \in A$ let $(\alpha', V') \to' (\alpha, V)$ iff $\alpha' = \alpha$ and $V' \to V$. The exogenous space $\mathbf{U}$ and noise distribution $P$ of $\mathcal{M}_A$ are the same as those of $\mathcal{M}$, the exogenous parents sets $\{\mathbf{U}(V)\}_V$ are also identical, and the functions are $\{f_{(\alpha, V)}\}_{(\alpha, V)}$ defined as follows. For any $\mathbf{W} := \mathbf{w} \in A$, $V \in \mathbf{V}$, $\mathbf{p} \in \chi_{\mathbf{Pa}(V)}$, and

35th Conference on Neural Information Processing Systems (NeurIPS 2021).

$\mathbf{u} \in \chi_{\mathbf{U}(V)}$ let

$$f_{(\mathbf{W}:=\mathbf{w},V)}\big((\mathbf{W} := \mathbf{w}, \mathbf{p}), \mathbf{u}\big) = \begin{cases} \pi_V(\mathbf{w}), & V \in \mathbf{W} \\ f_V(\mathbf{p}, \mathbf{u}), & V \notin \mathbf{W} \end{cases}.$$

# B  Proofs from §3

*Remark on exact characterizations of $\mathfrak{S}_3$, $\mathfrak{S}_2$.* Rich probabilistic languages interpreted over $\mathfrak{S}_3$ and $\mathfrak{S}_2$ were axiomatized in [5]. This axiomatization, along with the atomless restriction, gives an exact characterization for the hierarchy sets. Standard form, defined below, gives an alternative characterization exhibiting each $\mathfrak{S}_3^{\prec}$ as a particular atomless probability space (Corollary B.1.1). For $\mathfrak{S}_2^{X \to Y}$ (or $\mathfrak{S}_2$ in the two-variable case) we need the characterization for the proof of the hierarchy separation result, so it is given explicitly as Lemma B.3.1 in the section below on 2VE-spaces.  $\square$

## B.1  Standard Form

Fix $\prec$. Note that the map $\varpi_3$ restricted to $\mathfrak{M}_{\prec}$ does *not* inject into $\mathfrak{S}_3^{\prec}$, as any trivial reparametrizations of exogenous noise are distinguished in $\mathfrak{M}_{\prec}$. It is therefore useful to identify a "standard" subclass $\mathfrak{M}_{\prec}^{\mathrm{std}}$ on which $\varpi_3$ is injective with image $\mathfrak{S}_3^{\prec}$, and in which we lose no expressivity.

**Notation.** Let $\mathbf{Pred}(V) = \{V' : V' \prec V\}$ and denote a *deterministic* mechanism for $V$ mapping a valuation of its predecessors to a value as $\mathtt{f}_V \in \chi_{\mathbf{Pred}(V)} \to \chi_V$. Write an entire collection of such mechanisms, one for each variable, as $\mathtt{f} = \{\mathtt{f}_V\}_V$. A set $\mathbf{B} \subset \mathbf{V}$ is *ancestrally closed* if $\mathbf{B} = \bigcup_{V \in \mathbf{B}} \mathbf{Pred}(V)$. For any ancestrally closed $\mathbf{B}$ let $\xi(\mathbf{B}) = \big\{(V, \mathbf{p}) : V \in \mathbf{B}, \mathbf{p} \in \chi_{\mathbf{Pred}(V)}\big\}$. Note that $\mathrm{F}(\mathbf{B}) = \times_{(V,\mathbf{p}) \in \xi(\mathbf{B})} \chi_V$ encodes the set of all possible such collections of deterministic mechanisms, and we write, e.g., $\mathtt{f} \in \mathrm{F}(\mathbf{B})$. Abbreviate $\xi(\mathbf{V})$, $\mathrm{F}(\mathbf{V})$ for the entire endogenous variable set $\mathbf{V}$ as $\xi$, $\mathrm{F}$ respectively. We also use $\mathtt{f}$ to abbreviate the set

$$\bigcap_{\substack{V \in \mathbf{B} \\ \mathbf{p} \in \chi_{\mathbf{Pred}(V)}}} \pi^{-1}_{(\mathbf{Pred}(V):=\mathbf{p},V)}(\{\mathtt{f}(\mathbf{p})\}) \in \mathcal{B}(\chi_{A \times \mathbf{V}}) \tag{B.1}$$

so we can write, e.g., $p_{\mathrm{cf}}^{\mathcal{M}}(\mathtt{f})$ for the probability in $\mathcal{M}$ that the effective mechanisms $\mathtt{f}$ have been selected (by exogenous factors) for the variables $\mathbf{B}$.

**Definition B.1.1.** The SCM $\mathcal{M} = \langle \mathbf{U}, \mathbf{V}, \{f_V\}_V, P \rangle$ of Def. 3 is *standard form* over $\prec$, and we write $\mathcal{M} \in \mathfrak{M}_{\prec}^{\mathrm{std}}$, if we have that $\to \, = \, \prec$ for its influence relation, $\mathbf{U} = \{U\}$ for a single exogenous variable $U$ with $\chi_U = \mathrm{F}$, $P \in \mathfrak{P}(\mathrm{F})$ for its exogenous noise space, and for every $V$, we have that $\mathbf{U}(V) = \mathbf{U} = \{U\}$ and the mechanism $f_V$ takes $\mathbf{p}, (\{\mathtt{f}_V\}_V) \mapsto \mathtt{f}_V(\mathbf{p})$ for each $\mathbf{p} \in \chi_{\mathbf{Pred}(V)}$ and joint collection of deterministic functions $\{\mathtt{f}_V\}_V \in \mathrm{F} = \chi_U$.

Each unit $\mathbf{u}$ in a standard form model amounts to a collection $\{\mathtt{f}_V\}_V$ of deterministic mechanisms, and each variable is determined by a mechanism specified by the "selector" endogenous variable $U$.

**Lemma B.1.1.** Let $\mathcal{M} \in \mathfrak{M}_{\prec}$. Then there exists $\mathcal{M}^{\mathrm{std}} \in \mathfrak{M}_{\prec}^{\mathrm{std}}$ such that $\varpi_3(\mathcal{M}) = \varpi_3(\mathcal{M}^{\mathrm{std}})$.

*Proof.* To give $\mathcal{M}^{\mathrm{std}}$ define a measure $P \in \mathfrak{P}(\mathrm{F})$ as in Def. B.1.1 on a basis of cylinder sets by the counterfactual in $\mathcal{M}$

$$P\big(\pi^{-1}_{(V_1, \mathbf{p}_1)}(\{v_1\}) \cap \cdots \cap \pi^{-1}_{(V_n, \mathbf{p}_n)}(\{v_n\})\big)$$
$$= p_{\mathrm{cf}}^{\mathcal{M}}\big(\pi^{-1}_{(\mathbf{Pred}(V_1):=\mathbf{p}_1,V_1)}(\{v_1\}) \cap \cdots \cap \pi^{-1}_{(\mathbf{Pred}(V_n):=\mathbf{p}_n,V_n)}(\{v_n\})\big). \tag{B.2}$$

To show that $\varpi_3(\mathcal{M}) = \varpi_3(\mathcal{M}^{\mathrm{std}})$ it suffices to show that any two models agreeing on all counterfactuals of the form (B.2) must agree on all counterfactuals in $A$. Suppose $\alpha_i \in A$, $V_i \in \mathbf{V}$, $v_i \in \chi_{V_i}$ for $i = 1, \ldots, n$. Let $\mathbf{B} = \bigcup_i \mathbf{Pred}(V_i)$ and given $\mathtt{f} = \{\mathtt{f}_V\}_V$, define $\mathtt{f}_V^{\mathbf{W}:=\mathbf{w}}$ to be a constant function mapping to $\pi_V(\mathbf{w})$ if $V \in \mathbf{W}$ and $\mathtt{f}_V^{\mathbf{W}:=\mathbf{w}} = \mathtt{f}_V$ otherwise. Write $\mathtt{f} \vDash V = v$ if $\pi_V(\mathbf{v}) = v$ for that $\mathbf{v} \in \chi_{\mathbf{V}}$ such that $\mathtt{f}_V\big(\pi_{\mathbf{Pred}(V)}(\mathbf{v})\big) = \pi_V(\mathbf{v})$ for all $V$. Finally, note that

$$\bigcap_{i=1}^n \pi^{-1}_{(\alpha_i, V_i)}(\{v_i\}) = \bigsqcup_{\substack{\{\mathtt{f}_V\}_{V \in \mathbf{B}} \in \mathrm{F}(\mathbf{B}) \\ \{\mathtt{f}_V^{\alpha_i}\}_{V \in \mathbf{B}} \vDash V_i = v_i \\ \text{for each } i}} \{\mathtt{f}_V\}_{V \in \mathbf{B}}$$

where each set in the finite disjoint union is of the form (B.1). Thus the measure of the left-hand side can be written as a sum of measures of such sets, which use only counterfactuals of the form (B.2), showing agreement of the measures (by Fact 1). $\qquad\square$

**Corollary B.1.1.** $\mathfrak{S}_3^\prec$ bijects with the set of atomless measures in $\mathfrak{P}(\mathrm{F})$, which we denote $\mathfrak{S}_{\mathrm{std}}^\prec$. We write the map as $\varpi_{\mathrm{std}}^\prec : \mathfrak{S}_3^\prec \to \mathfrak{S}_{\mathrm{std}}^\prec$. $\qquad\square$

Where the order $\prec$ is clear, the above result permits us to abuse notation, using e.g. $\mu$ to denote either an element of $\mathfrak{S}_3^\prec$ or its associated point $\varpi_{\mathrm{std}}^\prec(\mu)$ in $\mathfrak{S}_{\mathrm{std}}^\prec$. We will henceforth indulge in such abuse.

*Proof of Fact 4.* The follows easily from Lem. B.1.2 below, adapted from Suppes and Zanotti [9, Thm. 1]. This shows that every atomless distribution is generated by some SCM; furthermore, it can chosen so as to exhibit no causal effects whatsoever. $\qquad\square$

**Definition B.1.2.** Say that $\nu \in \mathfrak{P}(\mathrm{F}(\mathbf{V}))$ is *acausal* if $\nu(\pi_{(V,\mathbf{p})}^{-1}(\{v_1\}) \cap \pi_{(V,\mathbf{p}')}^{-1}(\{v_2\})) = 0$ for every $(V, \mathbf{p}), (V, \mathbf{p}') \in \xi$ and $v_1 \neq v_2 \in \chi_V$.

**Lemma B.1.2.** Let $\mu \in \mathfrak{P}(\chi_{\mathbf{V}})$ be atomless. Then there is a $\mathcal{M} \in \mathfrak{M}_\prec^{\mathrm{std}}$ (see Def. B.1.1) with an acausal noise distribution such that $\mu = (\varpi_1 \circ \varpi_2 \circ \varpi_3)(\mathcal{M})$.

*Proof.* Consider $\nu \in \mathfrak{P}(\mathrm{F}(\mathbf{V})) = \mathfrak{P}(\bigtimes_{(V,\mathbf{p})} \chi_V)$ determined on a basis as follows: $\nu(\pi_{(V_1,\mathbf{p}_1)}^{-1}(\{v_1\}) \cap \cdots \cap \pi_{(V_n,\mathbf{p}_n)}^{-1}(\{v_n\})) = \mu(\pi_{V_1}^{-1}(\{v_1\}) \cap \cdots \cap \pi_{V_n}^{-1}(\{v_n\}))$. This is clearly acausal and atomless. $\qquad\square$

## B.2 Proofs from §3.2

*Proof of Prop. 2 (Collapse set $\mathfrak{C}_1$ is empty).* Let $\mu \in \mathfrak{S}_1$ and $\nu \in \mathfrak{S}_{\mathrm{std}}^\prec$ with $(\varpi_1 \circ \varpi_2 \circ \varpi_{\mathrm{std}}^{-1})(\nu) = \mu$. By Lemma B.1.2 we may assume $\nu$ is acausal. Let $X$ be the first, and $Y$ the second variable with respect to $\prec$. Note there are $x^*, y^*$ such that $\mu(\pi_X^{-1}(\{x^*\}) \cap \pi_Y^{-1}(\{y^*\})) > 0$; let $x^\dagger \neq x^*$, $y^\dagger \neq y^*$. Consider $\nu'$ defined as follows where $F_3$ stands for any set of the form $\pi_{(V_1,\mathbf{p}_1)}^{-1}(\{v_1\}) \cap \cdots \cap \pi_{(V_n,\mathbf{p}_n)}^{-1}(\{v_n\}) \subset \mathrm{F}(\mathbf{V})$, for $V_i \in \mathbf{V}$, $\mathbf{p}_i \in \chi_{\mathbf{P}(V_i)}$, $v_i \in \chi_{V_i}$, and $F_1$ is the corresponding $\pi_{V_1}^{-1}(\{v_1\}) \cap \cdots \cap \pi_{V_n}^{-1}(\{v_n\}) \subset \chi_{\mathbf{V}}$.

$$\nu'\big(\pi_{(X,())}^{-1}(\{x\}) \cap \pi_{(Y,(x^*))}^{-1}(\{y_*\}) \cap \pi_{(Y,(x^\dagger))}^{-1}(\{y_\dagger\}) \cap F_3\big) =$$
$$\begin{cases} \mu\big(\pi_X^{-1}(\{x^*\}) \cap \pi_Y^{-1}(\{y^*\}) \cap F_1\big), & x = x^*, y_* = y^* \neq y_\dagger \\ 0, & x = x^*, y_* = y^\dagger \neq y_\dagger \\ 0, & x = x^*, y_* = y_\dagger = y^* \\ \mu\big(\pi_X^{-1}(\{x^*\}) \cap \pi_Y^{-1}(\{y^\dagger\}) \cap F_1\big), & x = x^*, y_* = y_\dagger = y^\dagger \\ \mu\big(\pi_X^{-1}(\{x^\dagger\}) \cap \pi_Y^{-1}(\{y\}) \cap F_1\big), & x = x^\dagger \end{cases}$$

We claim that $\mu = \mu'$ where $\mu' = (\varpi_1 \circ \varpi_2)(\nu')$; it suffices to show agreement on sets of the form $\pi_X^{-1}(\{x\}) \cap \pi_Y^{-1}(\{y\}) \cap F_1$. If $x = x^\dagger$ then the last case above occurs; if $x = x^*$ and $y = y^\dagger$ then we are in the fourth case; if $x = x^*$ and $y = y^*$ then exclusively the first case applies. In all cases the measures agree. Let $(\nu_\alpha)_\alpha = \varpi_2(\nu)$ and $(\nu'_\alpha)_\alpha = \varpi_2(\nu')$ be the Level 2 projections of $\nu$, $\nu'$ respectively. Note that $\nu_{X:=x^\dagger}(y^\dagger) < \nu'_{X:=x^\dagger}(y^\dagger)$. This shows that the standard-form measures $\nu, \nu'$ project down to different points in $\mathfrak{S}_2$ (in particular differing on the $Y$-marginal at the index corresponding to the intervention $X := x^\dagger$) while projecting to the same point in $\mathfrak{S}_1$. Thus $\mu \notin \mathfrak{C}_1$ and since $\mu$ was arbitrary, $\mathfrak{C}_1 = \varnothing$. $\qquad\square$

**Example 2** (Collapse set $\mathfrak{C}_2$ is nonempty). We present a $\mu \in \mathfrak{S}_{\mathrm{std}}^\prec$ for which $\varpi_2(\mu) \in \mathfrak{C}_2$. Let $\mathbf{S}_n \subset \mathbf{V}$ be the ancestrally closed (§B.1) set of the $n$ least variables with respect to $\prec$ and $X$ be the first variable with respect to $\prec$; thus, e.g., $\mathbf{S}_1 = \{X\}$. Where $\mathbf{f} = \{\mathbf{f}_V\}_{V \in \mathbf{S}_n} \in \mathrm{F}(\mathbf{S}_n)$, define $\mu(\mathbf{f}) = 0$ if there is any $V \in \mathbf{S}_n \setminus \{X\}$, $\mathbf{p} \neq (0, \ldots, 0) \in \chi_{\mathbf{Pred}(V)}$ such that $\mathbf{f}_V(\mathbf{p}) = 0$, and otherwise define $\mu(\mathbf{f}) = 1/2^n$. Note that this example is *monotonic* in the sense of [1, 7].

We claim $\mu' = \mu$ for any $\mu' \in \mathfrak{S}_{\mathrm{std}}^\prec$ projecting to the same Level 2, i.e., such that $\varpi_2(\mu') = \varpi_2(\mu)$; note that it suffices to consider only candidate counterexamples with order $\prec$ since $\varpi_2(\mu) \notin \mathfrak{S}_2^{\prec'}$

for any $\prec' \neq \prec$. It suffices to show that $\mu(\mathbf{f}) = \mu'(\mathbf{f})$ for any $n$ and $\mathbf{f} = \{\mathbf{f}_V\}_{V \in \mathbf{S}_n}$; recall that in the measures, $\mathbf{f}$ denotes a set of the form (B.1). Let $(\mu_\alpha)_\alpha = \varpi_2(\mu) \in \mathfrak{S}_2^\prec$ and $(\mu'_\alpha)_\alpha = \varpi_2(\mu')$, with $(\mu_\alpha)_\alpha = (\mu'_\alpha)_\alpha$. Since $\mu'_{\mathbf{Pred}(V):=\mathbf{p}}(\pi_V^{-1}(\{1\})) = 1$ for any $V \in \mathbf{S}_n \setminus \{X\}$, $\mathbf{p} \neq (0, \ldots, 0)$, probability bounds show $\mu'(\mathbf{f})$ vanishes unless $\mathbf{f}_V(\mathbf{p}) = 1$ for each such $\mathbf{p}$, in which case

$$\mu'(\mathbf{f}) = \mu'\Big( \bigcap_{i=1}^n \pi^{-1}_{(V_i, \{V_1, \ldots, V_{i-1}\} := (0, \ldots, 0))}(\{v_i\}) \Big) \tag{B.3}$$

for some $v_i \in \chi_{V_i}$, where we have labeled the elements of $\mathbf{S}_n$ as $V_1, \ldots, V_n$, with $V_1 \prec \cdots \prec V_n$. We claim this is reducible—again using probabilistic reasoning alone—to a linear combination of quantities fixed by $(\mu'_\alpha)_\alpha$, the Level 2 projection of $\mu'$, which is the same as the projection $(\mu_\alpha)_\alpha$ of $\mu$. This can be seen by an induction on the number $m = |M|$ where $M = \{i : v_i = 1\}$: note (B.3) becomes

$$\mu'\Big( \bigcap_{i \notin M} \pi^{-1}_{(V_i, \{V_1, \ldots, V_{i-1}\} := (0, \ldots, 0))}(\{0\}) \Big)$$
$$- \sum_{M' \subsetneq M} \mu'\Big( \bigcap_{i \notin M'} \pi^{-1}_{(V_i, \{V_1, \ldots, V_{i-1}\} := (0, \ldots, 0))}(\{0\}) \cap \bigcap_{i \in M'} \pi^{-1}_{(V_i, \{V_1, \ldots, V_{i-1}\} := (0, \ldots, 0))}(\{1\}) \Big)$$

and the inductive hypothesis implies each summand can be written in the sought form while the first term becomes $\mu'\big( \bigcap_{i \notin M} \pi^{-1}_{(V_i, ())}(\{0\}) \big) = \mu'_{()}\big( \bigcap_{i \notin M} \pi^{-1}_{V_1}(\{0\}) \big) = \mu_{()}\big( \bigcap_{i \notin M} \pi^{-1}_{V_1}(\{0\}) \big)$. Here $()$ abbreviates the empty intervention $\varnothing := ()$. Thus any Level 3 quantity reduces to Level 2, on which the two measures agree by hypothesis.

### B.3 Remarks on §3.3

**Lemma B.3.1.** Let $(\mu_\alpha)_\alpha \in \bigtimes_{\alpha \in A_2^{X \to Y}} \mathfrak{P}(\chi_{X,Y})$. Then $(\mu_\alpha)_\alpha \in \mathfrak{S}_2^{X \to Y}$ iff

$$\mu_{X:=x}(x) = 1 \tag{B.4}$$

for every $x \in \chi_X$ and

$$\mu_{X:=x}(y) \geq \mu_{()}(x, y) \tag{B.5}$$

for every $x \in \chi_X, y \in \chi_Y$. Here $x, y$ abbreviates the basic set $\pi_X^{-1}(\{x\}) \cap \pi_Y^{-1}(\{y\})$.

*Proof.* It is easy to see that (B.4), (B.5) hold for any $(\mu_\alpha)_\alpha$. For the converse, consider the two-variable model over endogenous $\mathbf{Z} = \{X, Y\}$ with $X \prec Y$; note that $|\mathbf{F}(\mathbf{Z})| = 8$. A result of Tian et al. [10] gives that this model is characterized exactly by (B.4), (B.5) so for any such $(\mu_\alpha)_\alpha$ there is a distribution on $\mathbf{F}(\mathbf{Z})$ such that this model induces $(\mu_\alpha)_\alpha$. It is straightforward to extend this distribution to an atomless measure on $\mathbf{F}(\mathbf{V})$. $\qquad\square$

## C Proofs from §4

*Proof of Prop. 4.* This amounts to the continuity of projections in product spaces and marginalizations in weak convergence spaces. The latter follows easily from results in §3.1.3 of [4] or [3]. $\qquad\square$

*Proof of Thm. 2.* We show how Theorem 3.2.1 of [4] can be applied to derive the result. Specifically, let $\Omega = \bigtimes_\alpha \chi_{\mathbf{V}}$. Let $\mathcal{I}$ be the usual clopen basis, and let $W$ be the set of Borel measures $\mu \in \mathfrak{P}(\Omega)$ that factor as a product $\mu = \times_\alpha \mu_\alpha$ where each $\mu_\alpha \in \mathfrak{S}_1$ and $(\mu_\alpha)_\alpha \in \mathfrak{S}_2$. This choice of $W$ corresponds exactly to our notion of experimental verifiability.

It remains to check that a set is open in $W$ iff the associated set is open in $\mathfrak{S}_2$ (homeomorphism). It suffices to show their convergence notions agree. Suppose $(\nu_n)_n$ is a sequence, each $\nu_n \in W$, converging to $\nu = \times_\alpha \mu_\alpha \in W$. We have for each $n$ that $\nu_n = \times_\alpha \mu_{n,\alpha}$ such that $(\mu_{n,\alpha})_\alpha \in \mathfrak{S}_2$. By Theorem 3.1.4 in [4], which is straightforwardly generalized to the infinite product, for each fixed $\alpha$ we have $(\mu_{n,\alpha})_n \Rightarrow \mu_\alpha$. This is exactly pointwise convergence in the product space $\mathfrak{S}_2$, and the same argument in reverse works for the converse. $\qquad\square$

# D   Proofs from §5

We will use the following result to categorize sets in the weak topology.

**Lemma D.0.1.** If $X \subset \vartheta$ is a basic clopen, the map $p_X : (\mathfrak{S}, \tau^{\mathrm{w}}) \to ([0,1], \tau)$ sending $\mu \mapsto \mu(X)$ is continuous and open (in its image), where $\tau$ is as usual on $[0,1] \subset \mathbb{R}$.

*Proof.* Continuous: the preimage of the basic open $(r_1, r_2) \cap p_X(\mathfrak{S})$ where $r_1, r_2 \in \mathbb{Q}$ is $\{\mu : \mu(X) > r_1\} \cap \{\mu : \mu(X) < r_2\} = \{\mu : \mu(X) > r_1\} \cap \{\mu : \mu(\vartheta \setminus X) > 1 - r_2\}$, a finite intersection of the subbasic sets (1) from §4. See also Kechris [6, Corollary 17.21].

Open: if $X = \varnothing$ or $\vartheta$, then $p_X(\mathfrak{S}) = \{0\}$ or $\{1\}$ resp., both open in themselves. Else $p_X(\mathfrak{S}) = [0,1]$; we show any $Z = p_X\big(\bigcap_{i=1}^n \{\mu : \mu(X_i) > r_i\}\big)$ is open. Consider a mutually disjoint, covering $\mathcal{D} = \big\{\bigcap_{i=0}^n Y_i : Y_0 \in \{X, \vartheta \setminus X\}, \text{ each } Y_i \in \{X_i, \vartheta \setminus X_i\}\big\}$ and space $\Delta = \{(\mu(D))_{D \in \mathcal{D}} : \mu \in \mathfrak{S}\} \subset \mathbb{R}^{2^{n+1}}$. Just as in the Lemma, we have $\mathsf{p}_S : \Delta \to [0,1]$, for each $S \subset \mathcal{D}$ taking $(\mu(D))_D \mapsto \sum_{D \in S} \mu(D)$. Note $Z = \mathsf{p}_{\{D:D \cap X \neq \varnothing\}}\big(\bigcap_{i=1}^n \mathsf{p}_{\{D:D \cap X_i \neq \varnothing\}}^{-1}((r_i, 1])\big)$ so it suffices to show $\mathsf{p}_S$ is continuous and open; this is straightforward. $\qquad\square$

*Full proof of Lem. 1.* We show a stronger result, namely that the complement of the good set is nowhere dense. By rearrangement and laws of probability we find that the second inequality in (2) is equivalent to

$$\mu_x(y') < \mu_{()}(x') + \mu_{()}(x, y')$$
$$1 - \mu_x(y) < \underbrace{\mu_{()}(x') + \mu_{()}(x)}_{1} - \mu_{()}(x, y)$$
$$\mu_x(y) > \mu_{()}(x, y).$$

Lemma B.3.1 then entails the non-strict analogues of all four inequalities in (2), (3) are met for any $(\mu_\alpha)_\alpha \in \mathfrak{S}_2^{X \to Y}$, so we show that converting each to an equality yields a nowhere dense set, whose finite union is also nowhere dense. Note that we have a continuous and surjective observational projection $\pi_{()} : \mathfrak{S}_2^{X \to Y} \to \mathfrak{P}(\chi_{\{X,Y\}})$, and the first inequality in (3) is met iff $(\mu_\alpha)_\alpha \in (p_{x,y} \circ \pi_{()})^{-1}(\{0\})$ where $p_{x,y}$ is the map from Lemma D.0.1 and $x, y$ denotes the set $\pi_X^{-1}(\{x\}) \cap \pi_Y^{-1}(\{y\}) \subset \chi_{\{X,Y\}}$. This is nowhere dense as it is the preimage of the nowhere dense set $\{0\} \subset [0,1]$ under a map which is continuous by Lemma D.0.1. The second inequality of (3) is wholly analogous after rearrangement.

As for (2), define a function $d : \mathfrak{S}_2^{X \to Y} \to [0,1]$ taking $(\mu_\alpha)_\alpha \mapsto \mu_{X:=x}(y') - \mu_{()}(x, y')$; this function $d$ is continuous by Lemma D.0.1 and the continuity of addition and projection. Note that the first inequality of (2) holds iff $d((\mu_\alpha)_\alpha) = 0$. For any $\mu \in \mathfrak{S}_3^X$ such that $(\varpi_2^{X \to Y} \circ \varpi_2)(\mu) = (\mu_\alpha)_\alpha$, note that $d((\mu_\alpha)_\alpha) = \mu(x', y'_x)$ where $x', y'_x$ abbreviates the basic set $\pi_{((),X)}^{-1}(\{x'\}) \cap \pi_{(X:=x,Y)}^{-1}(\{y'\}) \in \mathcal{B}(\chi_{A \times \mathbf{V}})$. Thus $d$ is surjective, so that $d^{-1}(\{0\})$ is nowhere dense since $\{0\} \subset [0,1]$ is nowhere dense. The second inequality in (2) is again totally analogous. $\qquad\square$

*Proof of Lem. 2.* Abbreviate $\mu_3$ as $\mu$, and without loss take $\mu \in \mathfrak{S}_{\mathrm{std}}^{\prec}$. Note that (2), (3) entail

$$0 < \mu(x', y'_x) < \mu(x'), \quad 0 < \mu(x', y'_{x'}) < \mu(x').$$

and therefore

$$0 < \mu\big(\pi_{((),X)}^{-1}(\{x'\}) \cap \pi_{(x^*,Y)}^{-1}(\{1\})\big) < \mu\big(\pi_{((),X)}^{-1}(\{x'\})\big)$$

for each $x^* \in \chi_X = \{0,1\}$. In turn this entails that there are some values $y_0, y_1 \in \{0,1\}$ such that $\mu(\Omega_1) > 0, \mu(\Omega_2) > 0$ where the disjoint sets $\{\Omega_i\}_i$ are defined as

$$\Omega_1 = \pi_{((),X)}^{-1}(\{x'\}) \cap \pi_{(X:=0,Y)}^{-1}(\{y_0\}) \cap \pi_{(X:=1,Y)}^{-1}(\{y_1\})$$
$$\Omega_2 = \pi_{((),X)}^{-1}(\{x'\}) \cap \pi_{(X:=0,Y)}^{-1}(\{y_0^\dagger\}) \cap \pi_{(X:=1,Y)}^{-1}(\{y_1^\dagger\})$$

where in the second line, $y_0^\dagger = 1 - y_0$ and $y_1^\dagger = 1 - y_1$. Note that for $i = 1, 2$ we have conditional measures $\mu_i(S_i) = \frac{\mu(S_i)}{\mu(\Omega_i)}$ for $S_i \in \mathcal{B}(\Omega_i)$; further, $\Omega_i$ is Polish, since each is clopen. This implies

$\Omega_i$ is a standard atomless (since $\mu$ is) probability space under $\mu_i$. By Kechris [6, Thm. 17.41], there are Borel isomorphisms $f_i : \Omega_i \hookrightarrow [0,1]$ pushing $\mu_i$ forward to Lebesgue measure $\lambda$, i.e., $\mu_i(f_i^{-1}(B)) = \lambda(B)$ for $B \in \mathcal{B}([0,1])$. Thus $g = f_2^{-1} \circ f_1 : \Omega_1 \hookrightarrow \Omega_2$ is $\mu_i$-preserving: for $X_1 \in \mathcal{B}(\Omega_1)$,

$$\mu(g(X_1)) = \frac{\mu(\Omega_2)}{\mu(\Omega_1)}\mu(X_1). \tag{D.1}$$

Consider $\mu' = \varpi_3(\mathcal{M}')$ for a new $\mathcal{M}' \in \mathfrak{M}_\prec$, given as follows. Its exogenous valuation space is $\chi_{\mathbf{U}} = \Omega'$ where we define the sample space $\Omega' = \mathrm{F}(\mathbf{V}) \times \{\mathrm{T}, \mathrm{H}\}$; that is, a new exogenous variable representing a coin flip is added to some representation of the choice of deterministic standard form mechanisms. Fix constants $\varepsilon_1, \varepsilon_2 \in (0,1)$ with $\varepsilon_1 \cdot \mu(\Omega_1) = \varepsilon_2 \cdot \mu(\Omega_2)$ and define its exogenous noise distribution $P$ by

$$P(X \times \{\mathrm{S}\}) = \begin{cases} (1-\varepsilon_1) \cdot \mu(X), & X \subset \Omega_1, \mathrm{S} = \mathrm{T} \\ \varepsilon_1 \cdot \mu(X), & X \subset \Omega_1, \mathrm{S} = \mathrm{H} \\ (1-\varepsilon_2) \cdot \mu(X), & X \subset \Omega_2, \mathrm{S} = \mathrm{T} \\ \varepsilon_2 \cdot \mu(X), & X \subset \Omega_2, \mathrm{S} = \mathrm{H} \\ \mu(X), & X \subset \mathrm{F}(\mathbf{V}) \setminus (\Omega_1 \cup \Omega_2), \mathrm{S} = \mathrm{T} \\ 0, & X \subset \mathrm{F}(\mathbf{V}) \setminus (\Omega_1 \cup \Omega_2), \mathrm{S} = \mathrm{H} \end{cases} \tag{D.2}$$

Where $\mathtt{f} \in \mathrm{F}(\mathbf{V})$ and $V \in \mathbf{V}$ write $\mathtt{f}_V$ for the deterministic mechanism (of signature $\chi_{\mathbf{Pred}(V)} \to \chi_V$) for $V$ in $\mathtt{f}$. (Note that each $\mathtt{f}$ is just an indexed collection of such mechanisms $\mathtt{f}_V$.) The function $f'_V$ in $\mathcal{M}'$ is defined at the initial variable $X$ as $f'_X(\mathtt{f}, \mathrm{S}) = \mathtt{f}_X$ for both values of S, and for $V \neq X$ is defined as follows, where $\mathbf{p} \in \mathbf{Pred}(V)$:

$$f'_V(\mathbf{p}, (\mathtt{f}, \mathrm{S})) = \begin{cases} (g(\mathtt{f}))_V(\mathbf{p}), & \mathtt{f} \in \Omega_1, \mathrm{S} = \mathrm{H}, \pi_X(\mathbf{p}) = x \\ (g^{-1}(\mathtt{f}))_V(\mathbf{p}), & \mathtt{f} \in \Omega_2, \mathrm{S} = \mathrm{H}, \pi_X(\mathbf{p}) = x \\ \mathtt{f}_V(\mathbf{p}), & \text{otherwise} \end{cases} \tag{D.3}$$

We claim that $\varpi_2(\mu') = \varpi_2(\mu)$. It suffices to show for any $\mathbf{Z} := \mathbf{z} \in A$ and $\mathbf{w} \in \chi_{\mathbf{W}}$, $\mathbf{W}$ finite, we have

$$\mu(\theta) = \mu'(\theta), \text{ where } \theta = \bigcap_{W \in \mathbf{W}} \pi_{(\mathbf{Z} := \mathbf{z}, W)}^{-1}(\{\pi_W(\mathbf{w})\}). \tag{D.4}$$

Assume $\pi_Z(\mathbf{w}) = \pi_Z(\mathbf{z})$ for every $Z \in \mathbf{Z} \cap \mathbf{W}$, since both sides of (D.4) trivially vanish otherwise. Where $\mathtt{f} \in \mathrm{F}(\mathbf{V})$ write, e.g., $\mathtt{f} \vDash \theta$ if $m^{\mathcal{M}_A}(\mathtt{f}) \in \theta$, where $\mathcal{M}$ is a standard form model (Def. B.1.1); for $\omega' \in \Omega'$ write $\omega' \vDash' \theta$ if $m^{\mathcal{M}'_A}(\omega') \in \theta$. By the last two cases of (D.3) we have

$$\mu'(\theta) = \sum_{\mathrm{S}=\mathrm{T},\mathrm{H}} P\big(\{(\mathtt{f}, \mathrm{S}) \in \Omega' : (\mathtt{f}, \mathrm{S}) \vDash' \theta\}\big)$$
$$= \mu\big(\{\mathtt{f} \in \mathrm{F}(\mathbf{V}) \setminus (\Omega_1 \cup \Omega_2) : \mathtt{f} \vDash \theta\}\big) + \sum_{\substack{\mathrm{S}=\mathrm{T},\mathrm{H} \\ i=1,2}} P\big(\{(\mathtt{f}, \mathrm{S}) \in \Omega' : \mathtt{f} \in \Omega_i, (\mathtt{f}, \mathrm{S}) \vDash' \theta\}\big).$$
$$\tag{D.5}$$

Applying the first four cases of (D.2) and the third case of (D.3), the second term of (D.5) becomes

$$\sum_i \Big[\varepsilon_i \cdot \mu\big(\{\mathtt{f} \in \Omega_i : (\mathtt{f}, \mathrm{H}) \vDash' \theta\}\big) + (1-\varepsilon_i) \cdot \mu\big(\{\mathtt{f} \in \Omega_i : \mathtt{f} \vDash \theta\}\big)\Big]. \tag{D.6}$$

Either $X \in \mathbf{Z}$ and $\pi_X(\mathbf{z}) = x$, or not. In the former case: defining $X_i = \{\mathtt{f} \in \Omega_i : \mathtt{f} \vDash \theta\}$ for each $i = 1, 2$, the first two cases of (D.3) yield that

$$\{\mathtt{f} \in \Omega_1 : (\mathtt{f}, \mathrm{H}) \vDash' \theta\} = \{\mathtt{f} \in \Omega_1 : g(\mathtt{f}) \vDash \theta\} = g^{-1}(X_2)$$
$$\{\mathtt{f} \in \Omega_2 : (\mathtt{f}, \mathrm{H}) \vDash' \theta\} = \{\mathtt{f} \in \Omega_2 : g^{-1}(\mathtt{f}) \vDash \theta\} = g(X_1). \tag{D.7}$$

Applying (D.7) and (D.1), (D.6) becomes

$$\varepsilon_1 \cdot \frac{\mu(\Omega_1)}{\mu(\Omega_2)} \cdot \mu(X_2) + (1-\varepsilon_1) \cdot \mu(X_1) + \varepsilon_2 \cdot \frac{\mu(\Omega_2)}{\mu(\Omega_1)} \cdot \mu(X_1) + (1-\varepsilon_2) \cdot \mu(X_2)$$
$$= \mu(X_1) + \mu(X_2), \tag{D.8}$$

the final cancellation by choice of $\varepsilon_1, \varepsilon_2$. In the latter case: since $m^{\mathcal{M}}(\mathtt{f}) \in \pi_X^{-1}(\{x'\})$ for any $\mathtt{f} \in \Omega_1 \cup \Omega_2$, the third case of (D.3) gives $\{\mathtt{f} \in \Omega_i : (\mathtt{f}, \mathrm{H}) \vDash' \theta\} = X_i$. Thus (D.6) becomes (D.8) in either case. Putting in (D.8) as the second term in (D.5), we find $\mu(\theta) = \mu'(\theta)$.

Now we claim $\mu(\zeta) \neq \mu'(\zeta)$ for $\zeta = \zeta_0 \cap \zeta_1$ where $\zeta_1 = \pi_{(X:=1,Y)}^{-1}(\{y_1\})$ and $\zeta_0 = \pi_{(X:=0,Y)}^{-1}(\{y_0\})$. We have

$$
\begin{aligned}
\mu'(\zeta) =& \mu\big(\{\mathtt{f} \in \Omega \setminus (\Omega_1 \cup \Omega_2) : \mathtt{f} \vDash \zeta\}\big) \\
&+ \sum_{i=1,2} \Big[ \varepsilon_i \cdot \mu\big(\{\mathtt{f} \in \Omega_i : (\mathtt{f}, \mathrm{H}) \vDash' \zeta\}\big) + (1 - \varepsilon_i) \cdot \mu\big(\{\mathtt{f} \in \Omega_i : \mathtt{f} \vDash \zeta\}\big) \Big]. \quad \text{(D.9)}
\end{aligned}
$$

First suppose that $x = 0$. If $\mathtt{f} \in \Omega_1$, then note that $(\mathtt{f}, \mathrm{H}) \vDash' \zeta_0$ iff $g(\mathtt{f}) \vDash \zeta_0$, but this is never so, since $g(\mathtt{f}) \in \Omega_2$. If $\mathtt{f} \in \Omega_2$, then $(\mathtt{f}, \mathrm{H}) \vDash' \zeta_1$ iff $\mathtt{f} \vDash \zeta_1$, which is never so again by choice of $\Omega_2$. If $x = 1$ then we find that $(\mathtt{f}, \mathrm{H}) \nvDash \zeta_1$ (if $\mathtt{f} \in \Omega_1$) and $(\mathtt{f}, \mathrm{H}) \nvDash \zeta_0$ (if $\mathtt{f} \in \Omega_2$). Thus $(\mathtt{f}, \mathrm{H}) \nvDash' \zeta$ for any $\mathtt{f} \in \Omega_1 \cup \Omega_2$ and (D.9) becomes

$$
\mu\big(\{\mathtt{f} \in \Omega : \mathtt{f} \vDash \zeta\}\big) - \sum_{i=1,2} \varepsilon_i \cdot \mu\big(\{\mathtt{f} \in \Omega_i : \mathtt{f} \vDash \zeta\}\big) = \mu\big(\{\mathtt{f} \in \Omega : \mathtt{f} \vDash \zeta\}\big) - \varepsilon_1 \cdot \mu(\Omega_1) < \mu(\zeta).
$$

It is straightforward to check (via casework on the values $y_0, y_1$) that $\mu$ and $\mu'$ disagree also on the PNS: $\mu(y_x, y'_{x'}) \neq \mu'(y_x, y'_{x'})$ as well as its converse. As for the probability of sufficiency (Definition 10), note that

$$
P(y_x \mid x', y') = \frac{P(y_x, x', y'_{x'}) + \overbrace{P(y_x, y'_x, x', x)}^{0}}{P(x', y')}
$$

and it is again easily seen (given the definition of the $\Omega_i$) that $\mu(y_x, x', y'_{x'}) \neq \mu'(y_x, x', y'_{x'})$ while the two measures agree on the denominator; similar reasoning shows disagreement on the probability of enablement, since

$$
P(y_x \mid y') = \frac{P(y_x, y'_{x'}, x') + \overbrace{P(y_x, y'_x, x)}^{0}}{P(y')}. \qquad \square
$$