# OpenReview forum: "A Topological Perspective on Causal Inference"
_NeurIPS.cc/2021/Conference — NeurIPS 2021 Poster_

### Official Review · Reviewer_tnjA · 2021-07-15

**Rating:** 8
**Confidence:** 3

**Summary:**

The paper introduces a topological learning-theoretic version of the causal hierarchy theorem (Bareinboim et al. 2020).
This result shows that associational, causal, and counterfactual quantities are not only distinct in a measure-theoretic sense but also in a topological sense. This, in turn, implies that causal assumptions allowing the inference of arbitrary counterfactual quantities from experimental data would be statistically unverifiable.
This result reinforces the notion that causal assumptions are needed to perform causal and counterfactual inference, as they cannot be accomplished solely from data. Such formal limit becomes even more important when considering the strong inclination of ML research towards finding methods that are successful at modeling the observed data but are blind to causal constraints not present in the data.

**Limitations And Societal Impact:**

Limitations: I believe the scope and limitations of the result have been clearly stated in the paper.
Societal impacts: None that I can immediately foresee for the result.

**Main Review:**

This work gives a new and complementary perspective to recent work in formalizing the notion of the Ladder of Causation or the Pearl's Causal Hierarchy (PCH). The idea is to formally show the intuition that causal and counterfactual inference is impossible without causal assumptions given as part of the analysis. While this paper discusses the Causal Hierarchy Theorem (CHT) proposed by Bareinboim et al., it provides an alternative, novel, and complementary result. Moreover, the authors extend the result to models with an infinite number of discrete observable variables.

Based on limited knowledge of topologies and learning theory, the definitions and results presented in the paper look sound and make intuitive sense. Furthermore, the implications and limitations of the results have been discussed thoroughly.

I find the paper is very well written and approachable for researchers in the causal inference and machine learning community. I found the reading enjoyable and easy to make sense of as it gently introduces the mathematical concepts required to understand the result.

The results in the paper seem informative and relevant for any research that has the goal of inferring counterfactual and causal effects from data. Although the result imposes limits to counterfactual inference, it motivates the study of the assumptions needed to perform such a task and for building intelligent systems capable of causal and counterfactual reasoning.

**Time Spent Reviewing:**

5

---

> ### Author Response · Authors · 2021-08-10
> **Reply to Reviewer tnjA**
>
> Thank you very much for the encouraging review!

---

### Official Review · Reviewer_rxqC · 2021-07-16

**Rating:** 6
**Confidence:** 3

**Summary:**

This paper proposes a series of topologies on the different spaces of probability spaces of structural causal model. The authors define certain causal projections on these spaces that should map higher levels of the causal hierarchy to lower levels. The non-injectivity of these projection maps is shown and illustrates, what is called, the collapse of the hierarchy. The authors show that for those probability spaces that collapse are topologically meager. Using the result that open sets in the weak topology corresponds to statistical hypothesis that can be verified, the authors show that the collapse can never be statistically verified, because of topological meagerness.


**Limitations And Societal Impact:**

I think that this paper has the potential for a strong impact due to the novel connection between statistical verifiability and SCMs. I strongly recommend to elaborate more on the intuition of the crucial concepts, which I think will increase the impact of this paper even more.


**Main Review:**

This paper takes a very interesting and unique approach by connecting the world of topology of statistical verifiability to that of structural causal modeling. They show the interesting result, that any condition that could infer counterfactual distributions from interventional distributions must be statistically unverifiable, however, I found the paper very hard to read and I was unable to verify many of the claims.

First of all, the authors use a lot of non-standard and imprecise notation and terminology. For example, why not use the more common terminology, such as a graph, instead of direct influence relation? The authors state that their two assumptions of well-foundedness and local finiteness generalize the recursiveness assumption, however, they never define what well-foundedness is. This raise questions downstream such as what the intuition about $\omega$-like is. The non-standard notation of using projection mappings and cylinder subsets makes the paper quite difficult to follow. This led to much more confusion, later on. For example, in understanding the causal projection and how this gives rise to Fact. 3. By coincidence, while checking the Supplementary Material, I found that Definition 6 is not at all so trivial to understand and the authors had to derive that it was a  well-defined definition, without mentioning this, or referring to it, in the main paper.

Secondly, many things are not motivated or explained enough, for example what is the intuition of Definition 8? Why do you only consider those SCMs? Or, how does the result on meagerness relate to Meek's strong completeness result?

Thirdly, the authors seem not to be aware of related literature and many citations are missing IMO. For example, most of the results about SCMs and the causal hierarchy are already proven in a more general measure-theoretic setting, where (1) no countable range of the variables and (2) no recursiveness is assumed (see e.g., [1]). The authors do not motivate why they choose a more restricted setting, where only endogenous variables are taken with a countable range or they only consider recursiveness. This makes it difficult to distinguish what is already known (such as Proposition 2 and 3) and what the particular contributions of this paper are.

At last, I think the overall readability can be clarified by being more specific, and avoid sentences like '... it is sensible to aim for results with "nearly universal" force' or 'Common wisdom ... ' [without adding a reference]. I think in general, the results of this paper are very interesting and valuable, but I think that the clarity of writing can be improved a lot.  Given the current state of the paper, it does not provide a clear enough exposition of the results for NeurIPS.

[1] S. Bongers, P. Forré, J. Peters, and J. M. Mooij. Foundations of structural causal models with cycles
and latent variables. arXiv.org preprint, arXiv:1611.06221v5 [stat.ME], 2021. URL https://arxiv.org/abs/1611.06221

Update: I've updated my score.


**Time Spent Reviewing:**

7h

---

> ### Author Response · Authors · 2021-08-10
> **Reply to Reviewer rxqC**
>
> We are grateful to the reviewer for their comments and suggestions.
>
> We respectfully disagree with the statement that "most of the results about SCMs and the causal hierarchy are already proven in a more general measure-theoretic setting." All of our main results (Lemma 2, Theorem 3, Corollary 1) are new. To our knowledge, the closest results would be those of Bareinboim et al. 2020, who earlier proved a measure-theoretic version of our Theorem 3. Importantly (as we describe in the paper), the measure-theoretic setting there is strictly *less* general, and the interpretation and empirical significance are quite different.
>
> With that said, we very much welcome pointers to relevant literature that we may have missed. The paper by Bongers et al. is no doubt a significant contribution to the causal modeling literature. Whereas the vast majority of work in the area presumes acyclicity, that paper shows how much of the basic theory of SCMs extends to larger classes (like coupled differential equations) that eschew this assumption. We want to stress, however, that our results are not implied by any results in that paper. For instance, while that paper elegantly presents examples of separation of the hierarchy, it does not consider the question of how typical these examples are. As highlighted in the discussion with Reviewer 3EhT, our methods even allow showing that every probability of causation witnesses separation *almost-everywhere*.
>
> While orthogonal to our focus here on issues of statistical learnability, we see no impediment to topologizing these extended spaces using our same methods. Indeed, as mentioned in our conclusion, we have already begun work on a version with continuous endogenous variables. Thanks to the reviewer's suggestion, we have added this additional dimension as an important avenue for future work.
>
> Concerning the presentation: we appreciate that there may be other readers for whom "well-foundedness," "$\omega$-likeness," and "cylinder subsets" are unfamiliar, so we will add definitions and further background on these concepts as well.

---

> > ### Comment · Reviewer_rxqC · 2021-08-27
> > **Response**
> >
> > Thank you for answering my questions and providing further clarification. I think this paper can potentially be very impactful when all the clarification and additional pointers are carried out. Although most results in this paper are new and novel, I think overall the presentation of this paper can be improved. The third point I was trying to make is that it was often quite difficult to see what parts in the paper are novel and what parts are already known (due to missing citations). For example, I think, Proposition 1 is not new and is already proven in a more general setting in e.g., Bongers, et al., or Definition 9 or Proposition 2 is that a new definition and proposition or taken from [3]? etc.. I think that by making explicitly clear to the reader what is novel and what is not can make this a stronger paper. Given the authors' willingness to improve the paper, I will increase my score to a 6.

---

> > > ### Author Response · Authors · 2021-08-28
> > > **Response**
> > >
> > > Thank you for the encouraging and helpful response! The points about presentation are well taken. We will be careful about this and in particular will add clarification for each of the definitions and results you mention.

---

### Official Review · Reviewer_3EhT · 2021-07-17

**Rating:** 7
**Confidence:** 4

**Summary:**

This paper presents a topological framework to theorize about causal inference, and defines a weak topology on each of three spaces of probability measures, corresponding to the three levels of Pearl's causal hierarchy. In this setup a topological version of the causal hierarchy theorem is derived, to the effect that collapse of the hierarchy is topologically negligible. An interesting learning theoretic interpretation of this result is also given, in the spirit of no-free-lunch theorems.

**Limitations And Societal Impact:**

Yes.

**Main Review:**

This is an interesting and solid theoretical paper, providing a general framework for examining causal inference from a topological perspective, as well as establishing a topological version of a fundamental result about the limit of causal inference. Among papers of comparable technicality, this one is fairly readable and often illuminating. I like the paper, though it is hard to predict how much attention it will draw in the field of causality, let alone the NeurIPS community in general. I won't be surprised if the paper ends up with limited impact; we have to wait to see whether this perspective will yield more positive results. Still, I think the current paper is a good one for NeurIPS to consider.

Regarding the main result of the current paper, I think the bit in Corollary 1 about PNS in particular is more interesting than Theorem 3. Collapse is rarely called for in practice anyway, as we typically are concerned with specific causal questions such as PNS rather than with all possible causal questions. It will be interesting to know how typical it is that a specific counterfactual query is like the PNS query. I suspect that it is very typical, but I wonder whether the authors have a more definite answer or insight to add on this matter.

**Time Spent Reviewing:**

3

---

> ### Author Response · Authors · 2021-08-10
> **Reply to Reviewer 3EhT**
>
> We are grateful to the reviewer for their thoughtful remarks and suggestions. We agree that it is an interesting question whether other counterfactual questions are *PNS-like* in the sense that they witness comeager separation of Levels 2 and 3, and we appreciate the reviewer's point that (complete) collapse is rarely called for in practice.
>
> In fact, prompted by the reviewer's very useful comments here, we have taken the opportunity to expand our result to all of the major *probabilities of causation* considered in the literature (e.g., by Pearl 1999). Specifically, this includes not just PNS, i.e., $P(y_x,y_{x'}')$ and its dual $P(y_x',y_{x'})$, but also:
> 1. Probability of necessity: $P(y_{x'}'\mid x,y)$
> 2. Probability of sufficiency: $P(y_{x}\mid x',y')$
> 3. Probability of disablement: $P(y_{x'}' \mid y)$
> 4. Probability of enablement: $P(y_x \mid y')$
>
> In fact, the very same counterexample constructed in the proof of our main lemma separates all of these quantities.
>
> In other words, there is nothing special about PNS from the topological perspective. Any of these important counterfactual queries suffices to witness comeager separation. We would very much like to thank the reviewer for inspiring us to sharpen the statement of our main result in this way, and we have included this refinement in the paper.

---

> > ### Comment · Reviewer_3EhT · 2021-08-31
> > **Thanks for the response**
> >
> > I am happy with the response and will make a positive recommendation.

---

### Official Review · Reviewer_Ny4n · 2021-07-21

**Rating:** 7
**Confidence:** 3

**Summary:**

The authors propose an alternative analysis of how causal inference at different levels (observational, interventional, counterfactual) can interrelate by proposing a topology on the space of structural causal models. The authors' results confirm and at times extend existing results in the literature from a different theoretical perspective.

**Limitations And Societal Impact:**

For limitations see my comments above. I think the authors' topic does not acutely require a discussion of its societal impact, at least in its current incarnation.

**Main Review:**

In spite of the Increasing interest in causality in machine learning, I believe that an increase in the theoretical and philosophical understanding of the concept is still called for, preferably from different mathematical perspectives.

Therefore, I think the authors' contribution is timely, and would be of interest to the conference audience both in terms of its immediate findings and its potential downstream implications in theory and practice. However I also believe that the paper requires more thorough discussion of the potential marginal benefits of such an approach.

I think that the paper is well-written and does a fairly good job of introducing the concepts it uses in making its theoretical points. However, this is not to say that it is an easy read: after spending effort to penetrate the authors' terminology and theoretical results, one cannot help but expect to see more discussion regarding why adopting such a perspective useful for improving our understanding of causality. Such a discussion could include which theoretical commitments historically and/or currently limited developments in causality, and how the current perspective could help resolve these issues (an obvious historical example would be Pearl's justification of the do-calculus). Discussing how such theoretical commitments shaped practical research, and what new questions would their framework facilitate would also be very beneficial. They could also discuss the significance of their current findings that result from their theoretical framework. Also, weak points of a topological perspective would be a good point of discussion: what currently existing results cannot be replicated by the authors' framework?

The authors should not be expected to provide a complete account for all of the above, but in its current form the paper's findings feel like a complete theoretical overhaul only to confirm and modestly extend previous results. Although this is not unworthy of attention by itself, I think authors' work deserves a more thorough discussion in comparison to previous theoretical and empirical work in the field.


**Time Spent Reviewing:**

8

---

> ### Author Response · Authors · 2021-08-10
> **Reply to Reviewer Ny4n**
>
> We thank the reviewer for their helpful and sympathetic comments. We see our work in this paper not so much as supplanting previous research, but rather as building on and complementing existing frameworks and contributions. We would therefore prefer not to think of this as a "complete theoretical overhaul," but instead as a new angle on problems and model classes that have already received copious attention in the literature. We share the reviewer's excitement about "potential downstream implications in theory and practice" of the topological approach, and we hope that the payoffs presented in the submission give sufficient evidence of promise.
>
> The point that these payoffs should be clearly and unambiguously advertised is well taken. The advantages of a specifically *topological* perspective on the subject are two-fold, one technical and one conceptual. The technical point is generality, for instance in dealing with infinite variable ranges (see also the discussion with Reviewer rxqC about even further generalization). The conceptual point we see as truly fundamental: by topologizing spaces of causal models we establish a new bridge between causal inference and statistical learning theory. The immediate benefit is a learning theoretic interpretation of the causal hierarchy theorem, to the effect that causal assumptions needed to bridge Levels 2 and 3 are statistically unverifiable in a very strong sense. While this result is certainly compatible with the existing literature, we do not believe it is a mere confirmation of previous results.
>
> Following the reviewer's suggestions, we will make use of part of the additional allotted page to clarify these points further.

---

> > ### Comment · Reviewer_Ny4n · 2021-08-29
> > **Thank you for the response**
> >
> > I thank the authors for their response, in which I believe that the concerns I raised were properly addressed. I raise my score and (still) recommend the acceptance of the paper.

---

### Official Review · Reviewer_V1rj · 2021-08-02

**Rating:** 7
**Confidence:** 2

**Summary:**

This paper studies the weak topology on three spaces of probability measures, each corresponding to a level of a causal hierarchy introduced by Pearl (Pearl and Mackenzie, 2018).  The three levels, in order of increasing expressivity, are described as observational, interventional and counterfactual; given a set $\bf{V}$ of variables (known as "endogenous" variables; these variables can influence each other and may also be influenced by a set of "exogenous" variables), the observational distribution is a joint probability distribution over the values of $\bf{V}$ fixed according to some given structural causal model, an interventional distribution on $\bf{V}$ is a distribution where the values of some finite subset of $\bf{V}$ are fixed and a counterfactual distribution is one where various combinations of interventions with respect to different finite subsets of $\bf{V}$ are permitted.  The main results are: (1) a topological characterization of the set of hypotheses in the second level of the causal hierarchy that are "experimentally verifiable", which means that there is a sequence of tests that almost surely converge in the limit on the true hypothesis while a type I error is incurred up to a fixed bound at each stage (the relevant distribution belonging to the second level of the hierarchy); (2) the set C_2 of all points in the second level of the causal hierarchy that have a unique inverse (with respect to a projection from the third level) forms a meager set, where the underlying topology is the weak topology; a similar result applies to the preimage of C_2 under projection from the third level of the hierarchy.

**Ethical Concerns:**

None.

**Limitations And Societal Impact:**

Yes.

**Main Review:**

I think this is a well-written paper (in terms of language and style) that provides an interesting and new perspective of causation.  Mathematically, the paper seems generally correct, although the notation at some places was not clear to me.  I suggest checking the consistency and accuracy of the notation more carefully, particularly in definitions where nested sets are involved; while this is not a major point, I think it affects readability (more details below).  I am not familiar with the literature on causation, so I am unable to independently judge the overall novelty of the paper's contributions.  Based on what is written in the paper, it seems that the main innovation is to apply Genin and Kelly's (2017) topological approach to statistical verifiability to study causal models, in particular the three levels of the causal hierarchy proposed by Pearl.


Comments/Suggestions/Questions:

- Page 4, Definition 6: On the right-hand side of the equation, the meanings of the expressions (m^{M_{X:=x}})^{-1}(y)...,(m^{M_{W:=w}})^{-1}(z) were not clear to me.  (I am typing in plain text due to problems with the tex code displaying properly.)  According to the definition of (m^{M_{X:=x}})^{-1}, each input to (m^{M_{X:=x}})^{-1} should be a value in $\chi_{\bf{V}}$.  However, it seems from the definition of $\bf{y}$ between lines 147 and 148 that $\bf{y}$ is a value in $\chi_{(\bf{X}:=\bf{x}) \times Y}$.  But then why are the sets $\chi_{\bf{V}}$ and $\chi_{(\bf{X}:=\bf{x}) \times Y}$ comparable? I think some clarification would be good here.

- Page 5, definition of $\varpi_2$: The use of the "marginalization" operator $\varsigma$ here was not clear to me.  According to the definition at the end of page 4, $\varsigma$ is taken with respect to a _subset_ of the indexed variables; so if $A$ is a single variable, then should $\varsigma$ not be taken with respect to a subset of {$A$} $\cup \bf{V}$?  As I understand, {$\alpha$} $\times \bf{V}$ is not a subset of {$A$} $\cup \bf{V}$.

- Page 5, line of equations just before line 176: Why is $\varpi(\mathcal{M})$ not equal to $p^{\mathcal{M}}_{cf}$ (counterfactual probability)?

- Page 5, Definition 9: In the definition $\varpi(\mu) = \pi_{\emptyset}(\mu)$, it appears that $\mu$ is used to denote an indexed family of probability measures.  I suggest using a different notation (e.g. $(\mu^{(\alpha)})_{\alpha \in A}$) for a family of measures while reserving $\mu$ for a single measure.

- Page 9, line 371: lanscape -> landscape

- Supplementary, page 1, line 5: in the left-hand side of the equation, should one take the projection onto $Pa(V)$ of $m^{\mathcal{M}}(\mathbf{u})$?

- Supplementary, page 2, line 36: In the definition of $\xi(\bf{B})$, should $V \in \bf{V}$ be $V \in \bf{B}$ instead?  Otherwise, how does the definition depend on $\bf{B}$?  Similarly, in line 37, should the subscript be $(V,\bf{p}) \in \xi(\bf{B})$?

- Supplementary, page 2, Equation (B.2): On the left-hand side, should one take the inverse projections $\pi^{-1}$ instead?

- Supplementary, page 2, Definition B.1.1: The definition of an SCM here doesn't seem consistent with that of Definition 3 in the main paper.  Specifically, in the original definition of a mechanism f_V, f_V maps the values of a set of parents together with a set of values of U(V) to a value of V.  In Definition B.1.1, however, the second argument of f_V is {f_V}_V, a collection of deterministic functions, but {f_V}_V is not a member of $\bf{U} := \mathcal{B}(F)$, which is a collection of sets of collections of deterministic functions.  Should the second argument be written instead as {{$f_V$}$_V$}, i.e. the singleton whose member is {$f_V$}$_V$?

- Supplementary, page 3, Lemma B.1.2: I could not find the definition of an acausal model in the paper.

- Supplementary, page 3, line 74: I did not see in the proof of Proposition 2 where $\epsilon$ was used at all.

- Supplementary, page 3, last sentence of proof of Proposition 2: Did you mean that "$\mu' \notin C_1$ for every $\mu' \in G_1$ since $\mu$ was arbitrary"?

**Time Spent Reviewing:**

18

---

> ### Author Response · Authors · 2021-08-10
> **Reply to Reviewer V1rj**
>
> We thank the reviewer for the very careful and helpful comments!
> Below we respond to a selection of them. Where we have not responded, we have simply taken the reviewer's suggestion and fixed the typo or other minor issue in the paper.
>
> * Page 4, Definition 6: we "overload" boldface lowercase letters like $\mathbf{y}$ so that they refer both to members of $\chi_{\\{\mathbf{X} := \mathbf{x}\\} \times \mathbf{Y}}$, which usage the reviewer mentions, and to cylinder subsets of $\chi_{\mathbf{V}}$. The latter cylinder set is the set of all valuations for all the variables $\mathbf{V}$ that agree with $\mathbf{y}$ on the particular subset of variables $\mathbf{Y}$. The latter usage is defined formally in Definition 1, and we intended this usage in this particular instance. We have added a reference back to Definition 1 for clarity.
> * Page 5, definition of $\varpi_2$: marginalization is not always on specific endogenous variables, but on any index set. The counterfactual distribution is over a product space indexed by $A \times \mathbf{V}$ (i.e., pairs of interventions and variables). For any particular $\alpha \in A$, the set $\\{\alpha\\} \times \mathbf{V}$ is a subset of $A \times \mathbf{V}$; it contains exactly those pairs whose first element is $\alpha$.
> * Supplementary, page 2, Definition B.1.1: we should instead have a single exogenous variable $\textbf{U} = \\{U\\}$, each endogenous variable $V$ having it as a parent $\textbf{U}(V) = \textbf{U} = \\{U\\}$ and $\chi_U = \texttt{F}$. Thus a specific unit $\textbf{u}$ is just a collection $\\{\texttt{f}_V\\}_V$ of deterministic mechanisms. Each variable then takes on the mechanism that is specified by the "selector" exogenous variable $U$. We have corrected this in the paper.
> * Supplementary, page 3, Lemma B.1.2: acausal standard-form distribution is defined on page 2, Definition B.1.2. Here we mean the standard form model (Definition B.1.1) associated with the acausal distribution; indeed the goal of the standard form is just to exhibit the class of models as a set of probability distributions. We have clarified this in the statement of Lemma B.1.2.
> * Supplementary, page 3, line 74: indeed $\varepsilon$ is not needed, and we have removed it.
>
> We greatly appreciate the reviewer's scrupulous attention to detail.

---

### Author Response · Authors · 2021-08-10
**General Response**

We would like to thank all of the reviewers for the thoughtful, encouraging, and constructively helpful reviews. From their suggestions we have identified several concrete ways to improve the paper further. These are detailed in the individual responses, but in summary the main changes are:

1. Most substantially, a sharpening in the statement of the main theorem, showing that all of the *probabilities of causation* (not just PNS) witness comeager separation (thanks to Reviewer 3EhT).
2. A number of clarifications about specific terminology and notation (thanks to Reviewers V1rj and rxqC).
3. Further clarification concerning the specific advantages of the topological approach, also including more discussion of how it fits into the broader literature with which we engage (thanks to Reviewer Ny4n).
4. An additional pointer in the concluding remarks about future work, in which we could also explore spaces including *non-well-founded* (e.g., cyclic) causal models (thanks to Reviewer rxqC).

---

### Decision · Program_Chairs · 2021-09-27

**Decision:**

Accept (Poster)

**Comment:**

This paper proposes a novel topological approach to the causal hierarchy. All of the reviewers found the approach interesting, although some valid concerns were raised regarding clarity and presentation. In particular, a serious concern regarding comparisons with related work was raised and sorted out during the discussion phase.

In the end, there was a consensus recommendation to accept this paper which I concur with. We expect the authors will take the reviewer feedback into account, and in particular add a reference and detailed comparison with the paper [1].

[1] S. Bongers, P. Forré, J. Peters, and J. M. Mooij. Foundations of structural causal models with cycles and latent variables. arXiv.org preprint, arXiv:1611.06221v5 [stat.ME], 2021. URL https://arxiv.org/abs/1611.06221